# A meta-analysis of genome-wide association studies of childhood wheezing phenotypes identifies *ANXA1* as a susceptibility locus for persistent wheezing

Raquel Granell[1]*, John A Curtin[2], Sadia Haider[3], Negusse Tadesse Kitaba[4], Sara A Mathie[3], Lisa G Gregory[3], Laura L Yates[3], Mauro Tutino[2], Jenny Hankinson[2], Mauro Perretti[5], Judith M Vonk[6,7], Hasan S Arshad[8,9,10], Paul Cullinan[3], Sara Fontanella[3], Graham C Roberts[4,8,9], Gerard H Koppelman[7,11], Angela Simpson[2], Steve W Turner[12], Clare S Murray[2], Clare M Lloyd[3], John W Holloway[4,8], Adnan Custovic[3]*, on behalf of UNICORN and Breathing Together investigators

[1]MRC Integrative Epidemiology Unit, Department of Population Health Sciences, Bristol Medical School, University of Bristol, Bristol, United Kingdom; [2]Division of Infection, Immunity and Respiratory Medicine, School of Biological Sciences, The University of Manchester, Manchester Academic Health Science Centre, and Manchester University NHS Foundation Trust, Manchester, United Kingdom; [3]National Heart and Lung Institute, Imperial College London, London, United Kingdom; [4]Human Development and Health, Faculty of Medicine, University of Southampton, Southampton, United Kingdom; [5]William Harvey Research Institute, Barts and The London School of Medicine Queen Mary University of London, London, United Kingdom; [6]Department of Epidemiology, University of Groningen, University Medical Center Groningen\, Groningen, Netherlands; [7]University of Groningen, University Medical Center Groningen, Groningen Research Institute for Asthma and COPD (GRIAC), Groningen, Netherlands; [8]NIHR Southampton Biomedical Research Centre, University Hospitals Southampton NHS Foundation Trust, Southampton, United Kingdom; [9]David Hide Asthma and Allergy Research Centre, Isle of Wight, United Kingdom; [10]Clinical and Experimental Sciences, Faculty of Medicine, University of Southampton, Southampton, United Kingdom; [11]Department of Pediatric Pulmonology and Pediatric Allergology, University of Groningen, University Medical Center Groningen, Beatrix Children's Hospital, Groningen, Netherlands; [12]Child Health, University of Aberdeen, Aberdeen, United Kingdom

*For correspondence:
Raquel.Granell@bristol.ac.uk
(RG);
a.custovic@imperial.ac.uk (AC)

## Abstract

**Background:** Many genes associated with asthma explain only a fraction of its heritability. Most genome-wide association studies (GWASs) used a broad definition of 'doctor-diagnosed asthma', thereby diluting genetic signals by not considering asthma heterogeneity. The objective of our study was to identify genetic associates of childhood wheezing phenotypes.

**Methods:** We conducted a novel multivariate GWAS meta-analysis of wheezing phenotypes jointly derived using unbiased analysis of data collected from birth to 18 years in 9568 individuals from five UK birth cohorts.

**Results:** Forty-four independent SNPs were associated with early-onset persistent, 25 with pre-school remitting, 33 with mid-childhood remitting, and 32 with late-onset wheeze. We identified a novel locus on chr9q21.13 (close to annexin 1 [*ANXA1*], p<6.7 × 10$^{-9}$), associated exclusively with early-onset persistent wheeze. We identified rs75260654 as the most likely causative single nucleotide polymorphism (SNP) using Promoter Capture Hi-C loops, and then showed that the risk allele (T) confers a reduction in *ANXA1* expression. Finally, in a murine model of house dust mite (HDM)-induced allergic airway disease, we demonstrated that anxa1 protein expression increased and anxa1 mRNA was significantly induced in lung tissue following HDM exposure. Using anxa1$^{-/-}$ deficient mice, we showed that loss of anxa1 results in heightened airway hyperreactivity and Th2 inflammation upon allergen challenge.

**Conclusions:** Targeting this pathway in persistent disease may represent an exciting therapeutic prospect.

**Funding:** UK Medical Research Council Programme Grant MR/S025340/1 and the Wellcome Trust Strategic Award (108818/15/Z) provided most of the funding for this study.

## Editor's evaluation

The study uses a novel meta-analysis approach coupled with endotype discovery in GWAS studies of childhood wheezing to identify ANXA1 as a susceptibility locus. Functional data strengthens the conclusions from a relatively small sample size by GWAS standards. This is representative of a way forward for efficient genetic discovery in deeply phenotyped complex diseases, where there is much unrecognised heterogeneity.

## Introduction

Asthma is a complex disorder caused by a variety of mechanisms which result in multiple clinical phenotypes (*Pavord et al., 2018*). It has a strong genetic component, and twin studies estimate its heritability to be ~60–70% (*Duffy et al., 1990*). 'Asthma genes' have been identified through a range of approaches, from candidate gene association studies (*Simpson et al., 2012*) and family-based genome-wide linkage analyses (*Daniels et al., 1996*) to genome-wide association studies (GWASs) (*Moffatt et al., 2007*; *Moffatt et al., 2010*; *Demenais et al., 2018*). The first asthma GWAS (2007) identified multiple markers on chromosome 17q21 associated with childhood onset asthma (*Moffatt et al., 2007*). A comprehensive review summarising the results of 42 GWASs of asthma and asthma-related traits has been published recently (*El-Husseini et al., 2020*). The most widely replicated locus is 17q12-21, followed by 6p21 (*HLA* region), 2q12 (*IL1RL1/IL18R1*), 5q22 (*TSLP*), and 9p24 (*IL33*) (*Kim and Ober, 2019*). Overall, the evidence suggests that multiple genes are underlying the association peaks (*Kim and Ober, 2019*).

However, despite undeniable successes, genetic studies of asthma have produced relatively heterogeneous results, and only a small proportion of the heritability is accounted for (*Ober and Yao, 2011*). One part of the explanation for the paucity of precise replication are numerous gene-environment interactions (*Custovic et al., 2012*). Another important consideration is asthma heterogeneity, in that asthma diagnosis comprises several conditions with distinct pathophysiology (*Custovic, 2020*; *Haider et al., 2022*), each potentially underpinned by different genetic associations (*Custovic et al., 2019*). However, in order to maximise sample size, most GWASs used a definition of 'doctor-diagnosed asthma' (*Aaron et al., 2017*). Such aggregated outcome definitions are imprecise (*Looijmans-van den Akker et al., 2016*) and phenotypically and mechanistically heterogeneous (*Robinson et al., 2021*), and this heterogeneity may dilute important genetic signals (*Custovic et al., 2019*).

One way of disaggregating asthma diagnosis is to use data-driven methods to derive subtypes in a hypothesis-neutral way (*Howard et al., 2015*). For example, we jointly modelled data on wheezing from birth to adolescence in five UK population-based birth cohorts and identified five distinct phenotypes (*Oksel et al., 2019a*). However, although latent modelling approaches have been instrumental in elucidating the heterogenous nature of childhood asthma diagnosis (*Haider et al., 2022*), there has

**eLife digest** Three-quarters of children hospitalized for wheezing or asthma symptoms are preschool-aged. Some will continue to experience breathing difficulties through childhood and adulthood. Others will undergo a complete resolution of their symptoms by the time they reach elementary school. The varied trajectories of young children with wheezing suggest that it is not a single disease. There are likely different genetic or environmental causes.

Despite these differences, wheezing treatments for young children are 'one size fits all.' Studying the genetic underpinnings of wheezing may lead to more customized treatment options.

Granell et al. studied the genetic architecture of different patterns of wheezing from infancy to adolescence. To do so, they used machine learning technology to analyze the genomes of 9,568 individuals, who participated in five studies in the United Kingdom from birth to age 18. The experiments found a new genetic variation in the ANXA1 gene linked with persistent wheezing starting in early childhood. By comparing mice with and without this gene, Granell et al. showed that the protein encoded by ANXA1 controls inflammation in the lungs in response to allergens. Animals lacking the protein develop worse lung inflammation after exposure to dust mite allergens.

Identifying a new gene linked to a specific subtype of wheezing might help scientists develop better strategies to diagnose, treat, and prevent asthma. More studies are needed on the role of the protein encoded by ANXA1 in reducing allergen-triggered lung inflammation to determine if this protein or therapies that boost its production may offer relief for chronic lung inflammation.

been little research into the genetic associations of phenotypes derived using data-driven methods. This is the first study to investigate the genetic architecture of wheezing phenotypes from infancy to adolescence, to identify genes specific to each phenotype and better understand the genetic heterogeneity between the disease class profiles.

## Materials and methods

### Study design, setting, participants, and data sources/measurement

The Study Team for Early Life Asthma Research (STELAR) consortium (*Custovic et al., 2015*) brings together five UK population-based birth cohorts: Avon Longitudinal Study of Parents and Children (ALSPAC) (*Golding et al., 2001*), Ashford (*Cullinan et al., 2004*) and Isle of Wight (IOW) (*Arshad et al., 2018*) cohorts, Manchester Asthma and Allergy Study (MAAS) (*Custovic et al., 2002*), and the Aberdeen Study of Eczema and Asthma to Observe the Effects of Nutrition (SEATON) (*Martindale et al., 2005*). All studies were approved by research ethics committees. See Appendix 1: Description of cohorts for more details. Informed consent was obtained from parents, and study subjects gave their assent/consent when applicable.

Validated questionnaires were completed on multiple occasions from infancy to adolescence (*Oksel et al., 2019a*). A list of variables, per cohort, is shown in *Appendix 1—table 1*, and the cohort-specific time points and sample sizes in *Appendix 1—table 2*. Data were harmonised and imported into Asthma eLab web-based knowledge management platform to facilitate joint analyses (*Custovic et al., 2015*).

### Definition of primary outcome (wheeze phenotypes from infancy to adolescence)

In the pooled analysis among 15,941 subjects with at least two observations on current wheeze, we used latent class analysis (LCA) to derive wheeze phenotypes from birth to age 18 years (*Oksel et al., 2019a*). A detailed description of the analysis is presented in *Oksel et al., 2019a*, and in Appendix 1: Definition of variables. A five-class solution was selected as the optimal model (*Oksel et al., 2019a*), and the classes (wheeze phenotypes) were labeled as: (1) *never/infrequent wheeze* (52.4%); (2) *early-onset pre-school remitting wheeze* (18.6%); (3) *early-onset middle-childhood remitting wheeze* (9.8%); (4) *early-onset persistent wheeze* (10.4%); and (5) *late-onset wheeze* (8.8%). These latent classes were used in the subsequent GWAS.

## Genotyping, imputation, and GWAS meta-analysis

Genotyping, quality control, imputation, and exclusions are described in Appendix 2: Genotyping and imputation. Analyses were performed independently in ALSPAC, MAAS, and the combined IOW-SEATON-Ashford (genotyped on the same platform, at the same time, and imputed together). We used SNPTEST v2.5.2 (*Marchini and Howie, 2010*) with a frequentist additive multinomial logistic regression model (-method newml), using the never/infrequent wheeze as the reference and without including any covariates. A meta-analysis of the three GWASs was performed using METAL (*Willer et al., 2010*) with a total of 8,057,852 SNPs. See Appendix 2: LD clumping, pre-selection, and gene annotation for more details.

## Post-GWASs

Our GWAS identified a novel locus in chr9q21 nearby *Annexin A1* (*ANXA1*), exclusively associated with early-onset persistent wheeze (see Results section). We therefore proceeded with studies to identify causal variants and explore the biological mechanisms underlying this locus (see Appendix 3: Post-GWAS: rs75260654 (*ANXA1*) for more details). To this end, we firstly identified the most likely causative SNP using Promoter Capture Hi-C (PCHi-C) loops. We then ascertained genotype effect on gene expression and assessed the potential biological function of *ANXA1* in asthma. Finally, we used a murine model of house dust mite (HDM)-induced allergic airway disease to investigate whether *ANXA1* was important in regulating immune responses to a clinically relevant aeroallergen and used knock-out mice to derive further in vivo functional data to support our GWAS finding.

# Results

## Participants and descriptive data

We included a total of 9568 subjects with European ancestry: ALSPAC, n=6833; MAAS, n=887; SEATON, n=548; Ashford, n=348; and IOW, n=952. Demographic characteristics of the participants in STELAR cohorts included in this analysis and a flowchart are shown in *Appendix 1—table 3* and *Appendix 1—figure 1*. Cohorts contain similar proportions of males (range 48–54%), maternal history of asthma (11–14%), maternal smoking (14–23%), (doctor-diagnosed) asthma ever during mid-childhood (16–24%) and adolescence (20–30%), current wheeze (12–20% mid-childhood, 9–25% adolescence), and current use of asthma medication (12–17% mid-childhood, 11–17% adolescence). Individuals with missing genetic data as well as related and non-European individuals were excluded. Comparison of included vs. excluded individuals across cohorts (per cohort and time point) is in Appendix 1 and *Appendix 1—table 4*.

## GWAS meta-analysis

We conducted three GWASs (ALSPAC, MAAS, IOW-SEATON-Ashford) in parallel and results were meta-analysed. The distribution of the minor allele frequencies was consistent across genotyped datasets (mean SD 0.01). A circular Manhattan plot and a QQ plot are shown in *Figure 1*, *Figure 1—figure supplement 1*. Some observed p-values were clearly more significant than expected under the null hypothesis, particularly for early-onset persistent wheeze, without an early separation of the expected from the observed which indicates low evidence of population stratification. We observed slight deflation of the meta-analysis p-values in our summary statistics. Genomic inflation factor ($\lambda$) for early-onset pre-school remitting = 0.96, early-onset mid-childhood remitting = 0.94, late-onset = 0.96, and early-onset persistent wheezing = 0.97. A total of 589 SNPs were associated with at least one phenotype with $p < 10^{-5}$. After clumping, we identified 134 independent SNPs uniquely associated with different phenotypes ($p < 10^{-5}$): of these, 44 were exclusively associated with early-onset persistent, 25 with early-onset pre-school remitting, 33 with early-onset mid-childhood remitting, and 32 with late-onset wheeze (*Appendix 2—table 1*). Scatter plots in *Figure 2*, *Figure 2—figure supplement 1* show the heterogeneity in the genetic profile of the wheeze phenotypes. The plots show that all signals were phenotype-specific at $p < 10^{-5}$ and only nominal associations were shared across wheezing phenotypes. More details on how these plots were derived can be found in Appendix 2: Heterogeneity scatter plots. For example, chr17q21 was identified as a top locus for early-onset persistent wheeze ($p = 5.42 \times 10^{-9}$), but some of the SNPs in this region were also associated with the early-onset mid-childhood remitting phenotype ($p < 10^{-4}$).

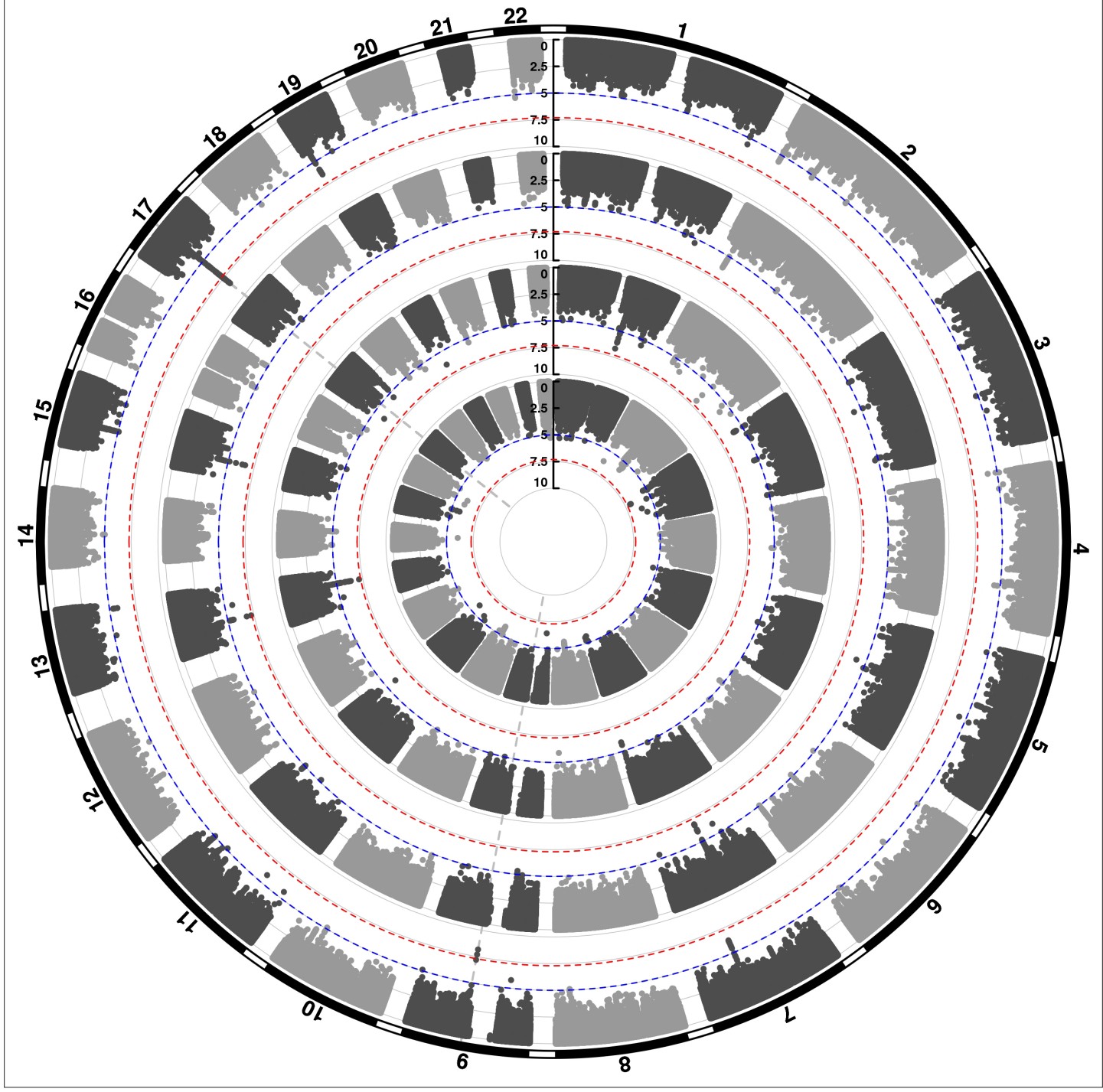

**Figure 1.** Circular Manhattan plot showing an overview of the genome-wide association study (GWAS) results by wheeze phenotype (from outside to inside: early-onset persistent, early-onset pre-school remitting, early-onset mid-childhood remitting, and late-onset wheeze). The red line indicates the genome-wide significance threshold (p < 5 × 10⁻⁸), while the blue line indicates the threshold for genetic variants that showed a suggestive significant association (p < 10⁻⁵).

The online version of this article includes the following figure supplement(s) for figure 1:

**Figure supplement 1.** QQ plots for each wheezing phenotype.

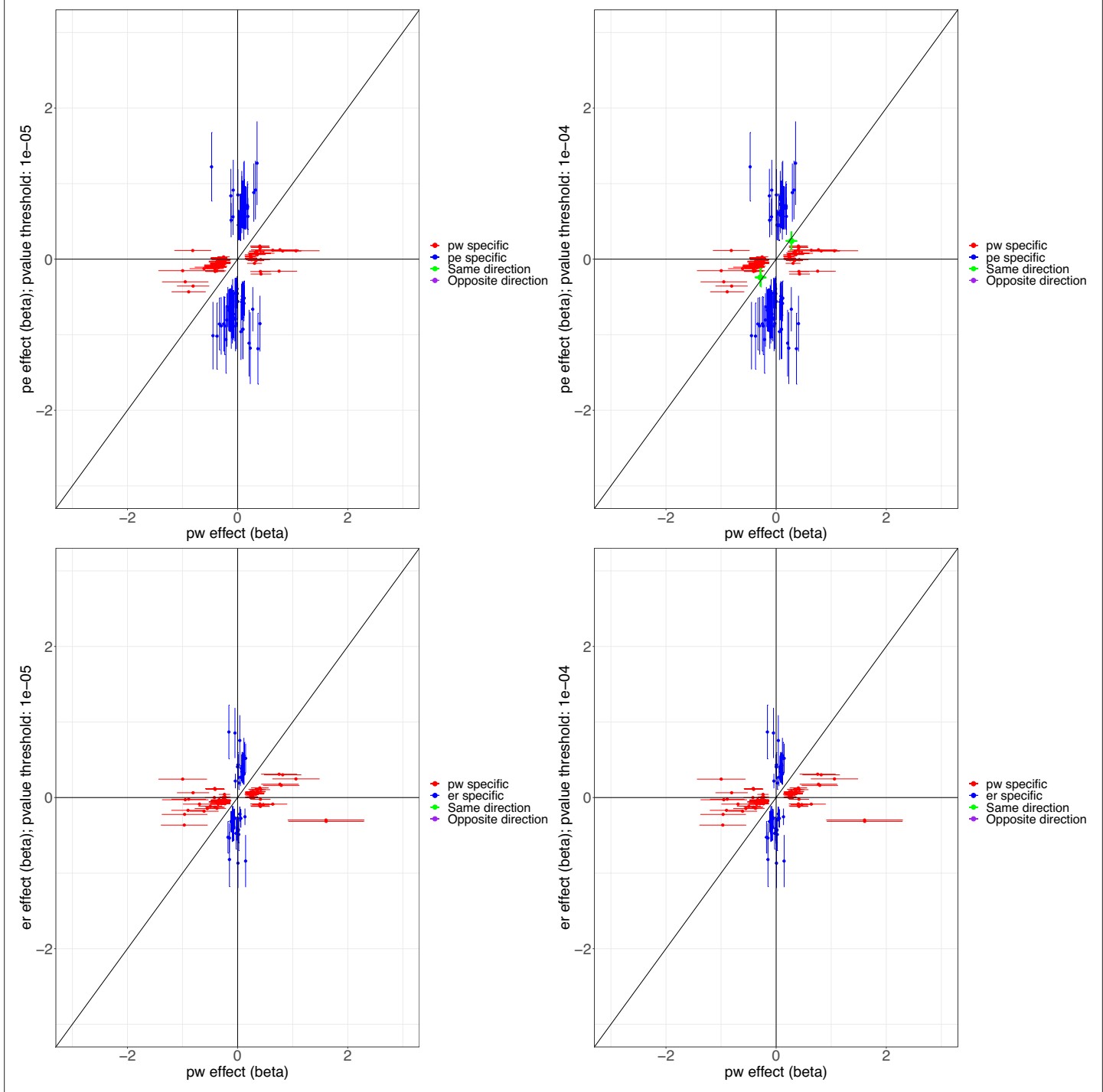

**Figure 2.** Scatter plots illustrating the heterogeneity in the genetic profile of the wheezing phenotypes. Top plots compare phenotype-specific beta effects for persistent and early-onset mid-childhood remitting wheezing. Shared nominal beta effects only found when relaxing $p < 10^{-4}$ for early-onset mid-childhood remitting wheezing. Bottom plots compare phenotype-specific beta effects for persistent and early-onset pre-school remitting. No shared beta effects (same or opposite direction) were found at $p < 10^{-5}$ for any of the comparisons. Abbreviations used: pw = persistent, er = early-onset pre-school remitting, and pe = early-onset mid-childhood remitting.

The online version of this article includes the following figure supplement(s) for figure 2:

**Figure supplement 1.** Scatter plots illustrating the heterogeneity in the genetic profile of wheezing phenotypes.

To help identify functional elements located near the GWAS-associated variants (potential causal variants), we used locus zoom plots (LZPs) for the 134 independent SNPs (p<10⁻⁵). Following close inspection of all plots, we short-listed 85 independent SNPs (*Appendix 2—table 1*) for which the LZPs potentially indicated more than one causal variant (*Appendix 2—figures 1–4*) and followed them up for further annotation. The results of GWAS meta-analysis for these 85 SNPs with main associations across the four wheeze phenotypes are presented in *Table 1*. Previously associated traits for each region/gene associated with the different wheeze phenotypes are shown in *Appendix 4— tables 1–4* and results are summarised in Appendix 4: Results in context of literature. Briefly, one region (6q27) among the top hits for early-onset pre-school remitting wheeze was previously associated with asthma, but in the context of obesity with a nominal association with asthma and BMI (*Melén et al., 2010*). Another region/gene (3q26.31/*NAALADL2*) identified as top hit for early-onset pre-school remitting wheeze was reported as an associate of severe asthma exacerbations, but only at nominal level (*Herrera-Luis et al., 2021*). No regions/genes identified as top hits for early-onset mid-childhood remitting wheeze were found to have previous associations with asthma. Several genes/loci identified as top hits for late-onset wheeze were previously associated with asthma: *ACOXL* chr2q13 (later onset asthma and obesity; *Zhu et al., 2020*), *PRKAA2* chr1p32.2 (lymphocyte count and asthma susceptibility; *Cusanovich et al., 2012*), *CD200* 3q13.2 (adult onset non-allergic asthma; *Siroux et al., 2014*), *GIMAP* family 7q36.1 (autoimmune diabetes, asthma, allergy; *Heinonen et al., 2015*), 9p22.3 (asthma in <16 years of age; *Denham et al., 2008*), and *16p12.1* (asthma and rhino-conjunctivitis at 10–15 years; *Sottile et al., 2019*).

We identified two GWAS-significant loci for early-onset persistent wheeze: 17q21, p<5.5 × 10⁻⁹, and a novel locus on 9q21.13 (*ANXA1*), p<6.7 × 10⁻⁹. The *ANXA1* locus was the only GWAS-significant locus that had not previously been associated with asthma or atopic traits, with one previous study showing an association with FEV₁/FVC and bronchodilator response in smokers (*Lutz et al., 2015*). *ANXA1* is strongly expressed in bronchial mast cells and has anti-inflammatory properties (*Vieira Braga et al., 2019*), and may be involved in epithelial airway repair (*Leoni et al., 2015*; *Appendix 4— table 1*). We therefore followed up top SNPs from this locus.

## *ANXA1* locus and persistent wheeze

Two SNPs (rs75260654, the lead SNP, and rs116849664 located downstream of *ANXA1*) were associated with early-onset persistent wheeze at genome-wide significance (GWS), with an additional SNP rs78320984 almost reaching GWS (*Appendix 5—table 1*). These SNPs are in linkage disequilibrium (LD) with each other (*Appendix 5—figure 1*), but not with any other SNPs.

### Promoter Capture identifies rs75260654 as the most likely causative variant

To identify the most likely causative variant, we investigated the overlap of the SNPs with PCHi-C interactions involving the *ANXA1* promoter in CD4+ cells in MAAS cohort subjects. Of the three SNPs, only rs75260654 overlapped a region interacting with the *ANXA1* promoter (*Figure 3*). More-over, rs75260654 overlapped a *POLR2A* ChIP-seq peaks and an ATAC-seq peak and active enhancer in the type II pneumocyte-derived A549 cell line. This shows that rs75260654 is located in a region directly interacting with the *ANXA1* promoter and is transcriptionally active in relevant cell types.

Allele frequencies of rs75260654 (MAF = 0.02) across wheeze phenotypes are shown in *Appendix 5—table 2*. Two individuals (one in MAAS and one in ALSPAC) were homozygote for the minor allele (T), and both were in the early-onset persistent wheeze class. One subject reported current wheeze and asthma through childhood, with hospitalisations for lower respiratory tract infection in the first year of life confirmed in healthcare records. The second individual reported current wheezing at 1.5, 2.5, and 8–9 years and doctor-diagnosed asthma and the use of asthma medication at 8–9 years.

### Rs75260654: effect on genomic features

Variant Effect Predictor (VEP) prediction shows the SNP rs75260654 (C changed to T) to be located downstream of three protein-coding transcripts of *AXNA1* and overlapping the known regulatory region ID ENSR00000882742 on Chromosome 9: 73,173,001–73,173,200. This region is active in the GI tract, M2 macrophages, neural progenitor cells, and trophoblasts, but is repressed in T lymphocytes including CD4+ CD25+, Treg, and CD8+ cells.

**Table 1.** Genome-wide association study (GWAS) meta-analysis: short-listed 85 top independent single nucleotide polymorphisms (SNPs) across the four wheezing phenotypes.

**Early-onset persistent wheezing**

| Locus | Independent SNPs | Nearby genes (SNPnexus) | Effect allele (freq)/ other allele | Beta | SE | p-Value | Effect direction (3 GWAS) | min_pval _other* | Previous relevant associations† |
|---|---|---|---|---|---|---|---|---|---|
| 1q43 | rs4620530 | CHRM3 | g(0.56)/t | 0.25 | 0.05 | 2.45E-06 | +++ | 0.79 | FEV$_1$, FEV$_1$/FVC, asthma- high priority drug target |
| 2p25.1 | rs13398488 | RNF144A | g(0.29)/a | 0.25 | 0.05 | 2.18E-06 | +-- | 0.13 | Asthma, allergy, childhood onset asthma, allergic rhinitis |
| 2q12.2 | rs6543291 | FHL2 | c(0.4)/t | 0.23 | 0.05 | 6.97E-06 | +++ | 0.10 | Bronchial hyper-responsiveness, airway inflammation; novel gene associated with asthma severity in human |
| 3q21.3 | rs77655717 | EFCC1, RAB43, RAB7A | c(0.05)/t | 0.47 | 0.10 | 6.40E-06 | +++ | 0.39 | RAB43: response to bronchodilator, FEV/FEC ratio; RAB7A: eosinophil count |
| 4p16.3 | rs7680608$^{eQTL}$ | RNF212, IDUA, DGKQ, SLC26A1 | g(0.93)/c | –0.42 | 0.09 | 1.31E-06 | --- | 0.15 | |
| | rs77822621$^{eQTL}$ | | c(0.96)/t | –0.50 | 0.11 | 7.16E-06 | --- | 0.01 | 4p16: asthma |
| 4q31.21 | rs115228498 | INPP4B | c(0.02)/t | 0.79 | 0.17 | 2.70E-06 | +++ | 0.02 | Atopic asthma |
| 5p15.31 | rs116494115 | ADCY2 | g(0.01)/a | 0.75 | 0.17 | 6.49E-06 | +++ | 0.09 | Asthma×air pollution, childhood asthma |
| 7q22.3 | rs76871421 | CDHR3 | c(0.12)/t | 0.37 | 0.07 | 5.71E-07 | +++ | 0.22 | Childhood asthma |
| 9q21.13 | rs75260654 | ANXA1, TMC1, LOC101927258, ALDH1A1 | c(0.98)/t | –0.90 | 0.16 | **6.66E-09** | --- | 0.05 | ANXA1: FEV$_1$/FVC, response to bronchodilators in smokers, with anti-inflammatory properties, strongly expressed in bronchial mast cells and potentially involved in epithelial airway repair |
| | rs116849664 | | c(0.98)/t | –0.89 | 0.16 | **1.99E-08** | --- | 0.06 | |
| 10q24.2 | rs7088157 | LOXL4, R3HCC1L | g(0.5)/a | –0.23 | 0.05 | 7.34E-06 | --- | 0.26 | R3HCC1L: eosinophil count, atopic eczema, psoriasis, BMI |
| 11p15.4 | rs112474574 | TRIM5, TRIM6, TRIM22 | c(0.96)/t | –0.55 | 0.12 | 2.29E-06 | --- | 0.14 | Severe asthma and insulin resistance |
| 11q23.3 | rs116861530$^{eQTL}$ | SIK3 | g(0.94)/a | –0.42 | 0.09 | 9.07E-06 | --- | 0.01 | Triglycerides, glucose metabolism, eosinophil count |
| 14q22.1 | rs1105683 | KTN1 | c(0.07)/t | 0.41 | 0.09 | 9.15E-06 | +++ | 0.24 | Severe asthma |
| 5q13.3 | rs2202714$^{eQTL}$ | FAM227B | g(0.36)/a | 0.23 | 0.05 | 8.71E-06 | +++ | 0.01 | rs35251997 and FEV$_1$; FEV$_1$/FVC |
| 15q25.2 | rs117540214$^{eQTL}$ | ADAMTSL3 | g(0.06)/a | 0.42 | 0.10 | 9.82E-06 | +++ | 3.91E-03 | FEV$_1$/FVC |
| 17q12 | rs17676191 | IKZF3 | g(0.10)/a | 0.36 | 0.08 | 2.18E-06 | +++ | 3.06E-03 | |
| | rs79026872 | | c(0.03)/t | 0.64 | 0.13 | 2.08E-06 | +++ | 2.56E-03 | |
| 17q21 | rs4795400 | GSDMB | c(0.53)/t | 0.30 | 0.05 | **5.42E-09** | +++ | 1.96E-04 | Early-onset asthma, persistent wheezing (chr17q12-q21) |
| | rs1031460 | | g(0.50)/t | 0.27 | 0.05 | 8.71E-08 | +++ | 1.87E-04 | |
| | rs56199421 | | c(0.45)/t | –0.23 | 0.05 | 4.50E-06 | --- | 9.61E-04 | |
| | rs4795406 | LRRC3C | g(0.55)/c | –0.24 | 0.05 | 9.91E-07 | --- | 1.51E-03 | |
| | rs72832972 | | c(0.92)/t | –0.38 | 0.08 | 8.91E-06 | --- | 0.01 | |
| | rs4794821 | | c(0.47)/t | 0.27 | 0.05 | 9.43E-08 | +++ | 1.07E-03 | |
| | rs59843584 | GSDMA | c(0.78)/a | –0.31 | 0.06 | 6.38E-08 | --- | 6.63E-03 | |
| | rs4804311 | | g(0.08)/a | 0.42 | 0.09 | 9.65E-07 | +-+ | 0.05 | |
| | rs2013694 | | c(0.89)/t | –0.38 | 0.08 | 8.29E-07 | --+ | 0.39 | |
| 19p13.2 | rs73501545 | MARCH2, HNRNPM, MYO1F | g(0.16)/a | 0.31 | 0.07 | 8.39E-06 | +++ | 0.29 | Triglycerides, HDL cholesterol, metabolic syndrome; MYO1F: FEV$_1$ and FVC |
| | rs111644945 | | g(0.9)/a | –0.41 | 0.08 | 4.01E-07 | --- | 0.02 | |

*Table 1 continued on next page*

*Table 1 continued*

**Early-onset persistent wheezing**

| Locus | Independent SNPs | Nearby genes (SNPnexus) | Effect allele (freq)/ other allele | Beta | SE | p-Value | Effect direction (3 GWAS) | min_pval _other* | Previous relevant associations† |
|---|---|---|---|---|---|---|---|---|---|
| | rs5994170 | | g(0.4)/a | 0.23 | 0.05 | 4.95E-06 | +++ | 0.58 | Triglycerides, eosinophil count, and body height |
| 22q11.1 | rs34902370 | CECR5 | c(0.75)/t | –0.25 | 0.06 | 6.80E-06 | --- | 0.41 | |

**Early-onset pre-school remitting wheezing**

| Locus | SNP | Nearby genes (SNPnexus) | Coded(freq)/other allele | Beta | SE | p-Value | Direction | min_pval _other | Previous relevant associations |
|---|---|---|---|---|---|---|---|---|---|
| 1q32.3 | rs12730098[eQTL] | PPP2R5A | c(0.79)/t | –0.22 | 0.05 | 8.44E-06 | --- | 0.53 | Waist circumference and obesity |
| | rs2880066 | | t(0.09)/a | 0.32 | 0.07 | 4.34E-06 | +++ | 0.20 | Airway repair in non-atopic asthma |
| 2p24.2 | rs10180268 | FAM49A or CYRIA | c(0.06)/t | 0.43 | 0.09 | 6.56E-07 | +++ | 0.19 | |
| | rs3861377 | NLGN1 | g(0.89)/a | –0.28 | 0.06 | 7.75E-06 | --- | 0.28 | Smoking |
| 3q26.31 | rs10513743 | NAALADL2 | c(0.84)/t | –0.25 | 0.06 | 4.97E-06 | -+- | 0.06 | Exacerbations requiring hospitalisation in asthma-suggestive p-value |
| 5q13.3 | rs10075253 | SV2C | c(0.85)/t | –0.27 | 0.06 | 1.20E-06 | --- | 0.17 | BMI |
| 6q27 | rs2453395 | PDE10A | g(0.33)/a | 0.19 | 0.04 | 9.51E-06 | +++ | 0.01 | Birth weight; asthma and BMI |
| | rs4730561 | | g(0.36)/a | –0.20 | 0.04 | 6.78E-06 | --- | 0.13 | Allergic diseases and atopy, smoking, BMI, airway wall thickness |
| | rs73144976 | | g(0.97)/a | –0.47 | 0.11 | 9.41E-06 | --- | 0.26 | |
| 7q21.11 | rs67259321 | MAGI2 | c(0.06)/t | 0.43 | 0.08 | 1.65E-07 | +-+ | 0.76 | |
| 9p13.3 | rs10758259[eQTL] | C9orf24 | g(0.17)/a | –0.27 | 0.06 | 4.64E-06 | --- | 0.01 | Airway repair |
| 11q22.3 | rs72994149 | GUCY1A2 | c(0.84)/t | –0.24 | 0.05 | 8.33E-06 | -+- | 0.06 | Systolic blood pressure |
| | rs2872948 | | t(0.96)/a | –0.54 | 0.10 | 5.93E-08 | --- | 0.27 | Systolic blood pressure |
| 13q21.1 | rs73527654 | PRR20A/B/C/D/E | g(0.08)/a | 0.34 | 0.07 | 2.85E-06 | +++ | 0.41 | |
| | rs116966886 | | g(0.99)/a | –0.82 | 0.18 | 7.55E-06 | -+- | 0.57 | Smoking |
| 15q21.1 | rs117565527 | SEMA6D | g(0.99)/a | –0.87 | 0.17 | 2.38E-07 | -+- | 0.43 | |

**Early-onset mid-childhood remitting wheezing**

| Locus | SNP | Nearby genes (SNPnexus) | Coded(freq)/ other allele | Beta | SE | p-Value | Direction | min_pval _other | Previous relevant associations |
|---|---|---|---|---|---|---|---|---|---|
| | rs35725789 | | c(0.95)/a | –0.56 | 0.12 | 5.42E-06 | -+- | 0.01 | |
| | rs146141555 | | c(0.98)/t | –0.89 | 0.17 | 2.04E-07 | -+- | 0.08 | |
| 1q23.2 | rs146575092 | CADM3, FCER1A, MPTX1, OR10J1 | g(0.98)/a | –0.85 | 0.17 | 8.73E-07 | -+- | 0.07 | Neutrophil count, CRP |
| 2p22.3 | rs7595553 | MRPL50P1 | g(0.16)/c | –0.46 | 0.10 | 3.26E-06 | --- | 0.12 | PM 2.5 exposure level and global DNA methylation level |
| 3p25.3 | rs34315999[eQTL] | RAD18 | c(0.03)/t | 0.69 | 0.14 | 1.11E-06 | +++ | 0.14 | Atopy/SPT |
| 3q29 | rs146961758 | MRPL50P1, LSG1, TMEM44-AS1, TMEM44, ATP13A3 | t(0.05)/a | 0.57 | 0.12 | 6.01E-06 | +-+ | 0.11 | *3q29:* BMI *TMEM44-AS1, TMEM44, ATP13A3:* diastolic blood pressure; *LSG1:* BMI, eosinophil count |
| 4q24 | rs138794367 | SLC9B1 | c(0.99)/t | –1.02 | 0.22 | 5.47E-06 | --- | 0.13 | Eosinophil count, allergic rhinitis |
| 5q14.1 | rs115719402 | AP3B1 | g(0.96)/a | –0.60 | 0.13 | 7.20E-06 | --- | 0.06 | Vital capacity, BMI |

*Table 1 continued on next page*

*Table 1 continued*

**Early-onset mid-childhood remitting wheezing**

| Locus | SNP | Nearby genes (SNPnexus) | Coded(freq)/ other allele | Beta | SE | p-Value | Direction | min_pval _other | Previous relevant associations |
|---|---|---|---|---|---|---|---|---|---|
| | rs9602218 | | c(0.06)/a | 0.58 | 0.12 | 1.74E-06 | +-+ | 0.05 | |
| | rs61960366 | | g(0.97)/a | −0.79 | 0.15 | 7.09E-08 | -+- | 0.12 | |
| | rs74589927 | *RNU6-67P, SLITRK1* | g(0.02)/a | 0.73 | 0.16 | 3.78E-06 | +-+ | 0.02 | |
| | rs2210726 | *VENTXP2, UBE2D3P4, MTND4P1* | c(0.91)/t | −0.47 | 0.10 | 1.33E-06 | --- | 0.02 | *RNU6-67P/* rs976078: food allergy |
| 13q31.1 | rs4390476 | | c(0.08)/a | 0.46 | 0.10 | 8.81E-06 | +++ | 0.12 | |
| 14q24.2 | rs117443464 | *ZFYVE1* | g(0.95)/a | −0.57 | 0.12 | 4.68E-06 | --+ | 0.19 | LDL cholesterol and systolic blood pressure |
| 20p12.3-p12.2 | rs6077514 | *PLCB4* | c(0.88)/t | −0.39 | 0.09 | 4.03E-06 | --- | 0.43 | Neutrophil count |

**Late-onset wheezing**

| Locus | SNP | Nearby genes (SNPnexus) | Coded(freq)/ other allele | Beta | SE | p-Value | Direction | min_pval _other | Previous relevant associations |
|---|---|---|---|---|---|---|---|---|---|
| 1p36.13 | rs9439669 | *KLHDC7A* | t(0.82)/a | −0.34 | 0.07 | 5.15E-06 | --- | 0.31 | *1p36.13*: metabolic syndrome |
| 1p32.2 | rs2051039 | *PPAP2B, PRKAA2* | c(0.08)/t | 0.47 | 0.10 | 6.06E-06 | +-+ | 0.08 | *PRKAA2*: lymphocyte count and asthma susceptibility |
| 1p31.1 | rs72673642 | *HMGB1P18* | g(0.77)/a | −0.31 | 0.07 | 6.25E-06 | --- | 0.01 | Smoking, BMI |
| 2q13 | rs140983998 | *ACOXL, BUB1* | c(0.98)/t | −0.88 | 0.19 | 4.71E-06 | --- | 0.40 | *ACOXL*: later onset asthma and obesity |
| 2q14.3 | rs148008098 | *AMMECR1L* | c(0.96)/t | −0.69 | 0.15 | 3.41E-06 | --- | 0.01 | Body height, blood protein; growth, bone, and heart alterations |
| 3p24.2 | rs4072729 | *RARB* | c(0.03)/t | 0.61 | 0.13 | 4.20E-06 | +-+ | 0.23 | FEV1/FVC, adult lung function |
| 3q13.2 | rs145629570 | *KIAA2018, NAA50, SIDT1, CD200* | c(0.02)/t | 0.92 | 0.18 | 6.83E-07 | +++ | 0.10 | *SIDT1*: FEV1/FVC; *CD200*: adult-onset non-allergic asthma |
| 3q23 | rs113643470 | *TFDP2, XRN1* | c(0.98)/t | −0.91 | 0.19 | 1.68E-06 | --- | 0.03 | *XRN1*: eosinophil count; *3q23*: allergic disease and atopic sensitisation |
| 4p11 | rs17472015 | *SLAIN2, SLC10A4, FRYL* | c(0.01)/t | 1.00 | 0.23 | 9.49E-06 | +++ | 0.46 | *FRYL*: body height, age at menopause |
| | rs117660982 | *KRBA1, ZNF467* | g(0.97)/a | −0.74 | 0.16 | 7.63E-06 | -+- | 0.18 | Systolic blood pressure |
| | rs118027705 | *GIMAP family, AOC1* | c(0.97)/t | −0.77 | 0.17 | 6.48E-06 | -+- | 0.01 | *AOC1*: CV disease, smoking; *GIMAP* family: autoimmune diabetes, asthma, and allergy |
| 7q36.1 | rs139489493 | *LOC105375566* | c(0.98)/t | −0.95 | 0.20 | 2.28E-06 | --- | 0.03 | |
| 7q36.3 | rs144271668 | *PTPRN2* | c(0.01)/a | 0.88 | 0.19 | 2.91E-06 | +++ | 0.28 | Eczema |
| 8q21.3 | rs990182 | *LOC105375631* | t(0.42)/a | 0.28 | 0.06 | 2.57E-06 | +++ | 0.46 | *8q21.3*: type 1 diabetes |
| 9p22.3 | rs79110962 | *NFIB, ZDHHC21* | c(0.08)/t | 0.51 | 0.10 | 3.98E-07 | +++ | 0.05 | *9p22.3*: asthma (mean age <16 years) |
| 10q23.31 | rs7896106 | *SLC16A12, IFIT family, PANK1* | g(0.35)/t | 0.30 | 0.06 | 1.35E-06 | +++ | 0.05 | *SLC16A12*: Body height; *PANK1*: insulin |
| 11q23.3 | rs141958628 | *CBL, CCDC84, MCAM* | c(0.98)/t | −0.98 | 0.20 | 1.33E-06 | -+- | 0.27 | *CCDC84*: asthma, allergy |
| 15q15.3-q21.1 | rs139134265 | *SPG11, CTDSPL2* | g(0.02)/c | 0.87 | 0.20 | 9.11E-06 | +-+ | 0.13 | *CTDSPL2*: alcohol drinking |
| 15q25.2 | rs143862030 | *ADAMTSL3, GOLGA6L4, UBE2Q2P8* | c(0.04)/t | 0.64 | 0.13 | 1.65E-06 | +++ | 0.08 | *ADAMTSL3*: FEV1/FVC; lean mass |
| 16p13.3 | rs113390367 | *SSTR5-AS1, CACNA1H* | g(0.86)/a | −0.40 | 0.08 | 1.04E-06 | --- | 0.16 | *CACNA1H*: eosinophil count |

*Table 1 continued on next page*

*Table 1 continued*

**Late-onset wheezing**

| Locus | SNP | Nearby genes (SNPnexus) | Coded(freq)/ other allele | Beta | SE | p-Value | Direction | min_pval _other | Previous relevant associations |
|---|---|---|---|---|---|---|---|---|---|
| *16p12.1* | rs4788025 | *GSG1L* | g(0.46)/a | −0.30 | 0.06 | 7.99E-07 | --- | 0.19 | *16p12.1*: current asthma and rhino-conjunctivitis at 10–15 years |
| *22q13.32* | rs133498 | *FAM19A5 or TAFA5* | g(0.94)/a | −0.48 | 0.11 | 5.35E-06 | --- | 0.84 | Obesity and metabolic dysfunction |

eQTL: identified in expression analyses of whole blood and/or lung tissues using Genotype-Tissue Expression database (https://gtexportal.org) using the European reference panel. Bold p-values are genome-wide significant (p < 5 × 10⁻⁸).

*Minimum p-value across associations with the other three wheezing phenotypes, using the never/infrequent wheeze as the baseline phenotype.

†List of references or sources (GeneCards, GWAS Catalog, PhenoScanner) available in ***Appendix 5—tables 1–4***.

## Rs75260654: effect on gene expression

The effect of rs75260654 on the expression of nearby genes was investigated by browsing the eQTL GTEX data available in Ensembl. Compared to C, the T allele was found to reduce the expression of *ANXA1* in naïve B cells (effect size = −2.36795, p=0.01) and to increase expression in lymphoblasoid cell lines (effect size = 0.848856, pe = 0.001) (***Figure 4***). This SNP affects expression of the neighbouring gene *ALDH1A1* (aldehyde dehydrogenase-1 family member A1) (effect size = −2.40446, p=0.0039 in macrophages infected with *Salmonella*). In the eQTL catalogue, rs75260654 is identified

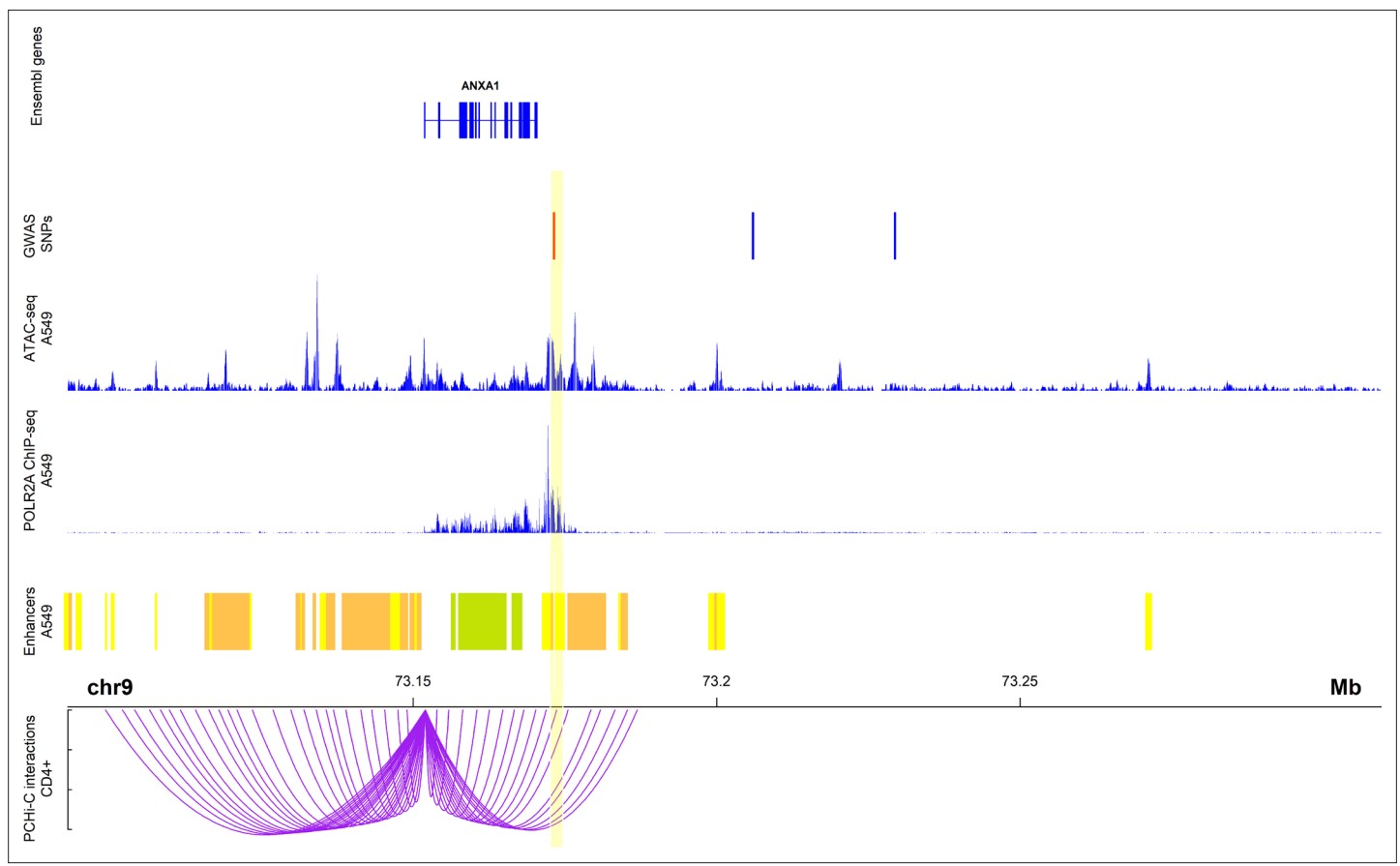

**Figure 3.** Chromatin interactions between rs75260654 and the *ANXA1* promoter in CD4+ cells in Manchester Asthma and Allergy Study (MAAS) *rs75260654* physically interacts with *ANXA1* promoter in CD4+ T cells and overlaps a region of active (POLR2AphosphoS2 ChIP-seq) open (ATAC-seq) chromatin in A549 cell line (lung epithelial carcinoma). The region is also predicted to be an active enhancer (ChromHMM 18-state model) in the A549 cell type. Only ChromHMM enhancer chromatin are displayed. Yellow shaded area indicates the Promoter Capture Hi-C (PCHi-C) fragment overlapping rs75260654 (red bar) and interacting with the *ANXA1* promoter.

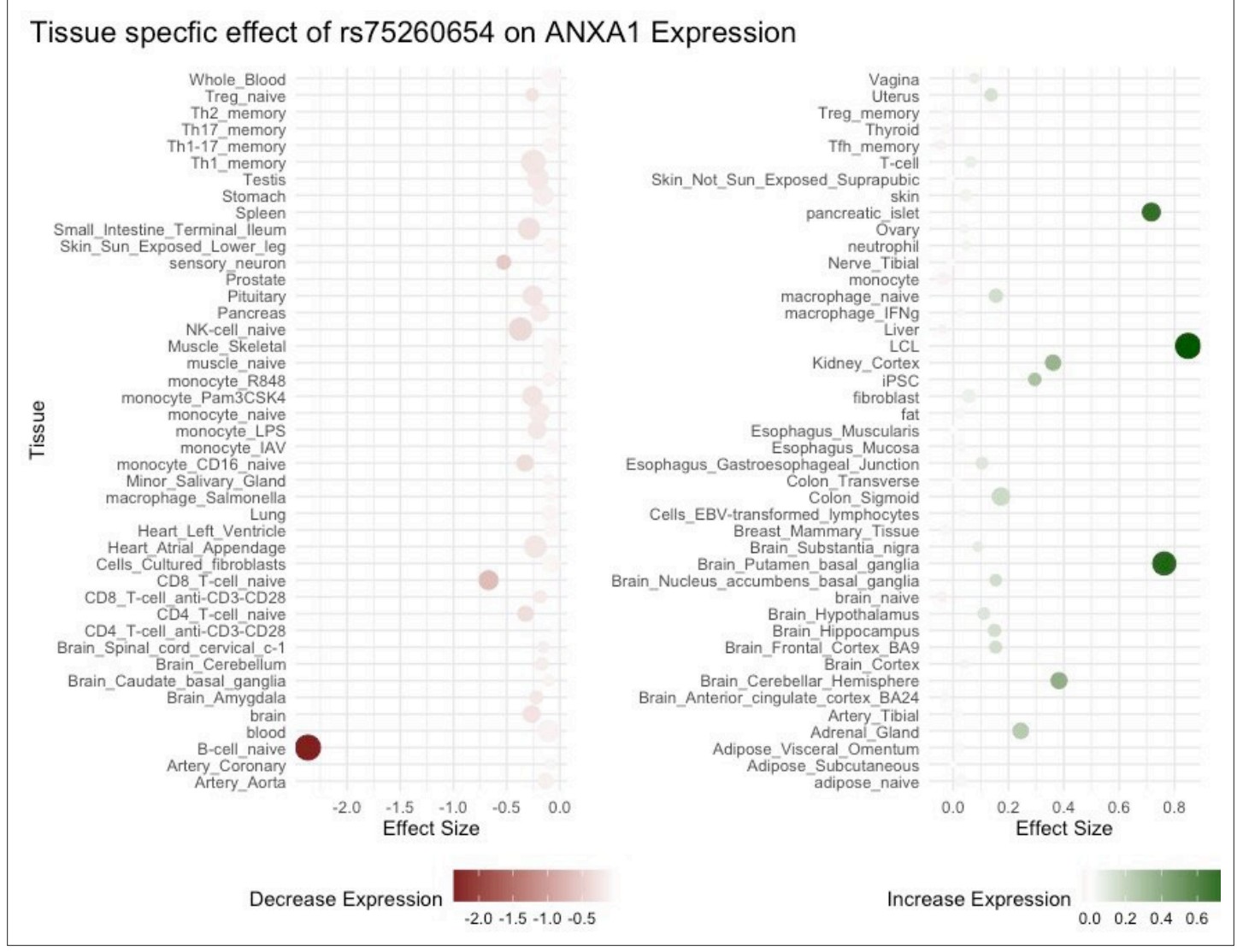

**Figure 4.** eQTL *ANXA1* and rs75260654 across different tissue types. Point size is proportional to -log10 p-value.

as an eQTL of *ANXA1* in various immune cells (at nominal significance) including T cells, monocytes, fibroblasts, whole blood, Th2 memory cells, naïve B cells. rs75260654 is also an eQTL of *ANXA1* in monocytes that were stimulated with R848 (agonist of TLRs 7 and 8) and a human seasonal influenza A virus (*Quach et al., 2016*) (at nominal significance) (*Appendix 5—table 3*). In the lung rs116849664 and rs78320984 (both in LD with rs75260654) were eQTLs of *ANXA1* (*Appendix 5—table 4*) as well as LINC01474 at nominal significance levels.

Additional supporting evidence regarding the significance of the T-allele on the expression of these genes was provided using eQTLGene Consortium meta-analysis of 24 cohorts and 24,331 samples (*Võsa et al., 2018*). This method reproduced the previous modest results showing a cis-eQTL effect of rs75260654 on both the *ANXA1* (p=6.02 × 10$^{-23}$) and *ALDH1A1* (p=1.11 × 10$^{-19}$) at FDR = 0. No significant trans-eQTLs were observed.

## Potential biological function of ANXA1 in asthma

Protein-protein network analysis demonstrated that *ANXA1* interacts directly with genes enriched for asthma (including *IL4* and *IL13*) and inflammatory regulation (*NR3C1*, glucocorticoid receptor) showing its significance in dysregulation of the immune response (see *Appendix 5—figure 2* and *Appendix 5—table 5*).

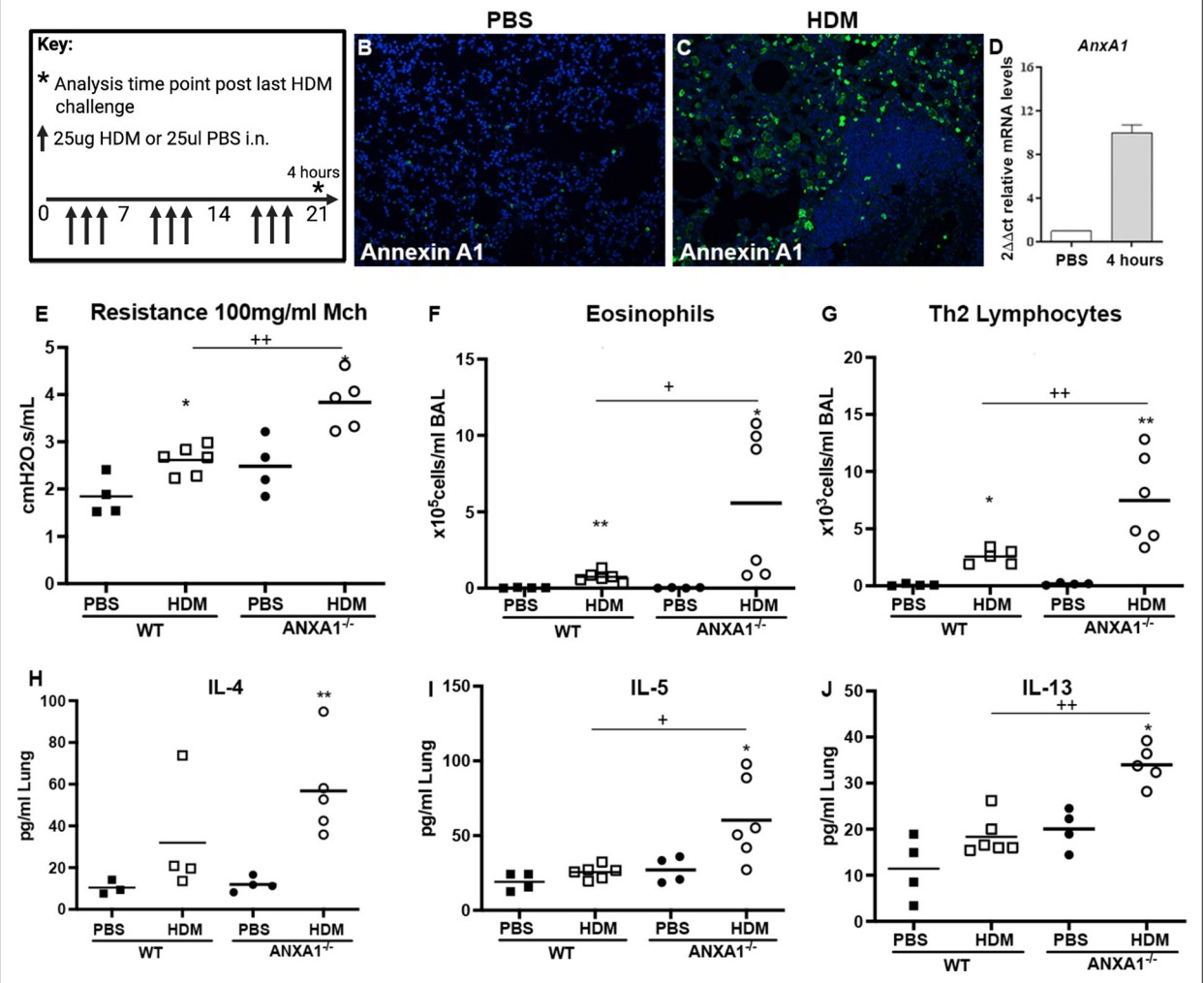

**Figure 5.** Annexin A1 is induced following house dust mite (HDM) challenge and mice deficient in *ANXA1* have exacerbated airway hyper-reactivity. (**A**) Schematic of HDM allergen dosing protocol, N=4–6 per group, data representative of two animal experiments. (**B, C**) Immunofluorescent staining of paraffin-embedded lung tissue sections incubated with anti-annexin A1, counterstained with DAPI (N=4 per group). (**D**) mRNA expression of annexin A1 in lung tissue following HDM exposure, expression normalised to housekeeping gene hprt. Mice receiving HDM were analysed for changes in airway hyper-reactivity following methacholine (MCh) challenge in tracheotomised restrained mice. (**E**) Airway resistance at top MCh dose 100 mg/ml. (**F**) Eosinophils quantified in BAL, (**F**) T1/ST2+ lymphocytes quantified in the BAL. (**H**) IL-4, (**I**) IL-5, and (**J**) IL-13 quantified in lung tissue by ELISA. *p<0.05 and **p<0.01 relative to PBS control group by Mann-Whitney test. +p < 0.05 and ++p < 0.01 comparing HDM annexin A1 knock-out (KO) mice relative to HDM wildtype (WT) group by Mann-Whitney test.

## Functional studies of *anxa1* in a murine model

### Pulmonary expression of anxa1 is modulated by aeroallergen exposure

We first analysed expression of *anxa1* using a model of HDM-induced allergic airway disease (*Figure 5A*; *Gregory et al., 2009*). Consistently, immunohistochemistry analysis revealed anxa1 protein expression increased following HDM challenge (*Figure 5B and C*). Anxa1 mRNA was significantly induced in lung tissue following HDM exposure (*Figure 5D*). This increase suggests that the pro-resolving *anxa1* may play a role in regulating the pulmonary immune response to allergen.

### Anxa1 suppresses allergen-induced airway hyperresponsiveness and type 2 inflammation

To confirm a functional role for *anxa1* in allergic airway disease, we exposed *anxa1−/−* mice to intranasal HDM. Wildtype (WT) mice given HDM over 3 weeks developed significant airway hyperresponsiveness (AHR) compared to PBS control mice. Mice deficient in *anxa1* had significantly worse lung

function (greater airway resistance) compared to WT-treated mice (*Figure 5E*). *Anxa1*[-/-] mice exhibited significantly increased airway eosinophilia and elevated numbers of Th2 lymphocytes (*Figure 5F and G*). Lung tissue cytokine levels reflected the exacerbated airway Th2 inflammation, with elevation in IL-4, and significant induction of IL-5 and IL-13 (*Figure 5H and J*). Thus, *anxa1* deficiency results in an alteration of the pulmonary immune response, with uncontrolled eosinophilia and an exacerbation of type 2 inflammation and AHR in response to allergen. More details in Appendix 6: Functional mouse experiments.

## Discussion

Herein, we present a comprehensive description of the genetic architecture of childhood wheezing disorders. Using a novel approach applied to a unique dataset from five UK birth cohorts, we identified subsets of SNPs differentially associated across four wheezing phenotypes: early-onset persistent (44 SNPs, 19 loci), early-onset pre-school remitting (25 SNPs, 10 loci), early-onset mid-childhood remitting (33 SNPs, 9 loci), and late-onset (32 SNPs, 20 loci). We found little evidence of genetic associations spanning across different phenotypes. This suggests that genetic architecture of different wheeze phenotypes comprises a limited number of variants likely underpinning mechanisms which are shared across phenotypes, but that each phenotype is also characterised by unique phenotype-specific genetic associations. Importantly, we identified a novel locus in chr9q21 nearby *ANXA1* exclusively associated with early-onset persistent wheeze ($p < 6.7 \times 10^{-9}$). To identify the most likely causative variant, we investigated the overlap of the associated SNPs with PCHi-C interactions to demonstrate that SNP rs75260654 overlapped a region interacting with the *ANXA1* promoter. Using eQTL data, we identified that the risk allele (T) of rs75260654 associated with early-onset persistent wheeze is also associated with *ANXA1* expression. Further investigation of the biological function of *ANXA1* revealed that it interacts with genes enriched for asthma (including *IL4* and *IL13*) and inflammatory regulation (*NR3C1*, glucocorticoid receptor). In functional mouse experiments, anxa1 protein expression increased and anxa1 mRNA was significantly induced in lung tissue following HDM exposure, suggesting that the pro-resolving anxa1 may play a role in regulating the pulmonary immune response to allergen. Concurrently, by utilising *anxa1*[-/-] deficient mice we demonstrated that loss of anxa1 results in heightened AHR and Th2 inflammation upon allergen challenge, providing important in vivo functional data to support our GWAS finding.

*ANXA1* is a 37 kDa glycoprotein with potent anti-inflammatory and pro-resolving properties that are mediated by interaction with a specific G protein-coupled receptor FPR2 (*Perretti et al., 2002*). This axis represents an important resolution pathway in chronic inflammatory settings such as those of rheumatoid arthritis (*D'Acquisto et al., 2008*) and ulcerative colitis (*Vong et al., 2012*). *ANXA1* belongs to the annexin family of Ca$^{2+}$-dependent phospholipid-binding proteins, and through inhibition of phospholipase A2, it reduces eicosanoid production, which also contributes to its anti-inflammatory activities. Modulation of M2 macrophage phenotype is also promoted by *ANXA1* to attenuate tissue inflammation (*McArthur et al., 2020*). Corticosteroids (a mainstay of asthma treatment) increase the synthesis of *ANXA1* (*Rhen and Cidlowski, 2005*). Plasma *ANXA1* levels are significantly lower in asthmatic patients with frequent exacerbations compared to those with stable disease, suggesting a link between this mediator and disease state (*Lee et al., 2018*). Furthermore, children with wheeze have reduced airway levels of *ANXA1* (*Eke Gungor et al., 2014*).

Previous functional studies using *anxa1*[-/-] deficient mice challenged with ovalbumin showed *anxa1*-deficient mice to have elevated AHR compared to WT mice (*Ng et al., 2011*). Ng et al. demonstrated that untreated *anxa1*-deficient mice have spontaneous AHR that predisposes them to exacerbated response to allergen (*Ng et al., 2011*). In the current study, we demonstrated in the murine lung the induction of Anxa1 in response to HDM exposure. In addition, genetic deletion of *anxa1* potentiated the development of AHR and enhanced eosinophilia and markers of Th2 inflammation in mice treated with HDM, which is consistent with and extends previous reports. Of interest, in mice, *anxa1* expression was recently found to be characteristic of a novel cell type called the Hillock cell, which may be involved in squamous barrier function and immunomodulation (*Montoro et al., 2018*). These data identify the ANXA1/FPR2 signalling axis as an important regulator of allergic disease, that could be manipulated for therapeutic benefit.

Our study has several limitations. By GWAS standards, our study is comparatively small and may be considered to be underpowered. The sample size may be an issue when using an aggregated

definition (such as 'doctor-diagnosed asthma') but is less likely to be an issue when primary outcome is determined by deep phenotyping. This is indirectly confirmed in our analyses. Our primary outcome was derived through careful phenotyping over a period of more than two decades in five independent birth cohorts, and although comparatively smaller than some asthma GWASs, our study proved to be powered enough to detect previously identified key associations (e.g., chr17q21 locus). Precise phenotyping has the potential to identify new risk loci. For example, a comparatively small GWAS (1173 cases and 2522 controls) which used a specific subtype of early-onset childhood asthma with recurrent severe exacerbations as an outcome identified a functional variant in a novel susceptibility gene *CDHR3* (SNP rs6967330) as an associate of this disease subtype, but not of doctor-diagnosed asthma (*Bønnelykke et al., 2014*). This important discovery was made with a considerably smaller sample size but using a more precise asthma subtype. In contrast, the largest asthma GWAS to date had an ~40-fold higher sample size (*Demenais et al., 2018*), but reported no significant association between *CDHR3* and aggregated asthma diagnosis. Therefore, with careful phenotyping, smaller sample sizes may be adequately powered to identify larger effect sizes than those in large GWASs with broader outcome definitions (*Schoettler et al., 2019*).

The importance of the precise outcome definition was highlighted in our previous studies in ALSPAC which explored genetic associates of wheeze phenotypes derived by LCA (*Granell et al., 2013*; *Spycher et al., 2012*). Our current findings are consistent with our earlier report suggesting that 17q21 SNPs are associated with early-onset persistent, but not with early transient or late-onset wheeze (*Granell et al., 2013*). Further analysis using genetic prediction scores based on 10–200,000 SNPs ranked according to their associations with physician-diagnosed asthma found that the 46 highest ranked SNPs predicted persistent wheeze more strongly than doctor-diagnosed asthma (*Spycher et al., 2012*). Finally, a candidate gene study combining data from ALSPAC and PIAMA found different associations of IL33-IL1RL1 pathway polymorphisms with different phenotypes (*Savenije et al., 2014*).

We are cognisant that there may be a perception of the lack of replication of our GWAS findings. We would argue that direct replication is almost certainly not possible in other cohorts, as phenotypes for replication studies should be homogenous (*Crawford et al., 2015*). However, there is a considerable heterogeneity in LCA-derived wheeze phenotypes between studies, and although phenotypes in different studies are usually designated with the same names, they differ between studies in temporal trajectories, distributions within a population, and associated risk factors (*Oksel et al., 2018*). This heterogeneity is in part consequent on the number and the non-uniformity of the time points used, and is likely one of the factors responsible for the lack of consistent associations of discovered phenotypes with risk factors reported in previous studies (*Oksel et al., 2019b*). This will also adversely impact the ability to identify phenotype-specific genetic associates. For example, we have previously shown that less distinct wheeze phenotypes in PIAMA were identified compared to those derived in ALSPAC (*Savenije et al., 2011*). Thus, phenotypes that are homogeneous to those in our study almost certainly cannot readily be derived in available populations. This is exemplified in our attempted replication of *ANXA1* findings in PIAMA cohort (see Appendix 5: Replication of *ANXA1* top hits in PIAMA cohort and *Appendix 5—table 6*). In this analysis, the number of individuals assigned to persistent wheezing in PIAMA was small (*Võsa et al., 2018*), associates of this phenotype differed to those in STELAR cohorts, and the SNPs' imputation scores were low (<0.60), which meant the conditions for replication were not met.

Our study population is of European descent, and we cannot generalise the results to different ethnicities or environments. It is important to highlight the under-representation of ethnically diverse populations in most GWASs (*Kim and Ober, 2019*). To mitigate against this, large consortia have been formed, which combine the results of multiple ethnically diverse GWASs to increase the overall power to identify asthma susceptibility loci. Examples include the GABRIEL (*Moffatt et al., 2010*), EVE (*Torgerson et al., 2011*), and TAGC (*Demenais et al., 2018*) consortia, and the value of diverse, multiethnic participants in large-scale genomic studies has recently been shown (*Wojcik et al., 2019*). However, such consortia do not have the depth of longitudinal data to allow the type of analyses which we carried out to derive a multivariable primary outcome. Finally, the manual and visual inspection of LZPs for the refinement of association signals and identification of functional elements was an objective approach which might have undermined the findings. One strength of our approach is that we used data from five birth cohorts with detailed and lifelong phenotyping, which were harmonised in a common knowledge management platform (*Custovic et al., 2015*), allowing joint analyses. We

performed three parallel GWASs that produced estimates with remarkably consistent directions of effects.

In conclusion, using unique data from five UK birth cohorts, we identified subsets of SNPs differentially associated across four wheezing phenotypes from infancy to adolescence. We found little evidence of genetic associations spanning across different phenotypes. We discovered a novel locus in chr9q21 uniquely associated with early-onset persistent wheeze (p<6.7 × 10$^{-9}$), identified SNP rs75260654 as the most likely causative variant, and demonstrated that the risk allele (T) confers a reduction in *ANXA1* expression. In mouse experiments, *ANXA1* expression increased in lung tissue following allergen exposure, suggesting that the pro-resolving ANXA1 may play a role in regulating the pulmonary immune response to allergen. Using *ANXA1*-deficient mice, we demonstrated that loss of *ANXA1* results in heightened AHR and Th2 inflammation upon allergen challenge, providing important in vivo functional data to support our GWAS finding. Targeting these pathways to promote the clearance of chronic inflammation in persistent disease may represent an exciting therapeutic prospect.

## Acknowledgements

STELAR/UNICORN investigators: Professor Graham Devereux, Dr Dimitrios Charalampopoulos. Breathing Together investigators: Prof Andrew Bush, Prof Sejal Saglani, Prof Benjamin Marsland, Prof Jonathan Grigg, Prof Jurgen Schwarze, Prof Mike Shields, Prof Peter Ghazal, Prof Ultan Power, Dr Ceyda Oksel.

Supported by the UK Medical Research Council (MRC) Programme Grant MR/S025340/1, the Wellcome Trust (WT) Strategic Award (108818/15/Z) and a WT Senior Fellowship to CML (107059/Z/15/Z). The MRC and Wellcome (Grant ref: 217065/Z/19/Z) and the University of Bristol provide core support for ALSPAC. ALSPAC GWAS data was generated by Sample Logistics and Genotyping Facilities at Wellcome Sanger Institute and LabCorp (Laboratory Corporation of America) using support from 23andMe. A comprehensive list of grants funding is available on the ALSPAC website (http://www.bristol.ac.uk/alspac/external/documents/grant-acknowledgements.pdf).

PIAMA was funded by the Netherlands Lung Foundation (grant 3.4.01.26, 3.2.06.022, 3.4.09.081 and 3.2.10.085CO), the ZON-MW Netherlands Organization for Health Research and Development (grant 912-03-031), the Stichting Astmabestrijding and the Ministry of the Environment. Genome-wide genotyping was funded by the European Commission as part of GABRIEL (grant number 018996) and a grant from BBMRI-NL (CP 29). GHK is supported by a ZON-MW VICI grant.

## Additional information

### Competing interests

Graham C Roberts: MRC grant to my institution; President of the British Society of Allergy and Clinical Immunology. Gerard H Koppelman: Dutch Lung Foundation, Ubbo Emmius Foundation (Money to insitition); Dutch Lung Foundation, Vertex, TEVA the Netherlands, GSK, ZON-MW (VICI grant), European Union (Money to institution); Astra Zeneca, Pure IMS, GSK (Money to institution); Sanofi, Boehringer Ingelheim (Money to institution). Angela Simpson: Medical research council Research grant; JP Moulton Charitable Foundation Research grant; Asthma UK Research grant. Clare S Murray: has received grants from Asthma Uk, the National Institute for Health Research, the Moulton Charitable Foundation and the North West Lung Centre Charity (to the Institution). They received lecture fees from GSK and Novartis, and received a travel grant from Sanofi. The authors has no other competing interests to declare. Clare M Lloyd: Wellcome Trust 107059/Z/15/Z. John W Holloway:

Medial Research Council grant MR/S025340/1 (to institution); American Academy of Allergy Asthma and Immunology (AAAI) (Support for speaker travel to AAAAI annual congress). Adnan Custovic: MRC (research grants); EPSRC (research grant); Wellcome Trust (research grant); Worg Pharmaceoticals (Personal payment). The other authors declare that no competing interests exist.

## Funding

| Funder | Grant reference number | Author |
|---|---|---|
| UK Medical Research Council | MR/S025340/1 | Raquel Granell<br>Adnan Custovic |
| Wellcome Trust | 108818/15/Z | Raquel Granell<br>Adnan Custovic |
| Wellcome Trust | 107059/Z/15/Z | Clare M Lloyd |

The funders had no role in study design, data collection and interpretation, or the decision to submit the work for publication. For the purpose of Open Access, the authors have applied a CC BY public copyright license to any Author Accepted Manuscript version arising from this submission.

## Author contributions

Raquel Granell, Conceptualization, Data curation, Formal analysis, Supervision, Methodology, Writing – original draft, Writing – review and editing; John A Curtin, Formal analysis, Visualization, Methodology, Writing – original draft, Writing – review and editing; Sadia Haider, Formal analysis, Writing – review and editing; Negusse Tadesse Kitaba, Writing – review and editing, Contributed to interpretation of results, Post-gwas analyses; Sara A Mathie, Writing – review and editing, Contributed to work related to mouse models; Lisa G Gregory, Writing – original draft, Writing – review and editing, Contributed to work related to mouse models; Laura L Yates, Writing – review and editing, Contributed to work related to mouse models; Mauro Tutino, Visualization, Writing – review and editing, Contributed to interpretation of results post-gwas; Jenny Hankinson, Writing – review and editing, Designed and carried out the HiC work; Mauro Perretti, Writing – review and editing, Contributed to work related to mouse models; Judith M Vonk, Writing – review and editing, Replication in PIAMA; Hasan S Arshad, Paul Cullinan, Sara Fontanella, Graham C Roberts, Angela Simpson, Steve W Turner, Clare S Murray, Writing – review and editing; Gerard H Koppelman, Writing – review and editing, Replication in PIAMA; Clare M Lloyd, Writing – review and editing, Contributed to work related to mouse models; John W Holloway, Conceptualization, Writing – review and editing; Adnan Custovic, Conceptualization, Resources, Supervision, Funding acquisition, Writing – original draft

## Author ORCIDs

Raquel Granell http://orcid.org/0000-0002-4890-4012
Negusse Tadesse Kitaba http://orcid.org/0000-0001-7518-9096
Mauro Perretti http://orcid.org/0000-0003-2068-3331
John W Holloway http://orcid.org/0000-0001-9998-0464
Adnan Custovic http://orcid.org/0000-0001-5218-7071

## Ethics

ALSPAC: Ethical approval for the study was obtained from the ALSPAC Ethics and Law Committee and the Local Research Ethics Committees. Informed consent for the use of data collected via questionnaires and clinics was obtained from participants following the recommendations of the ALSPAC Ethics and Law Committee at the time. All self-completion questionnaire content is approved by the ALSPAC Ethics and Law Committee. Bristol and Weston Health Authority: E1808 Children of the Nineties: Avon Longitudinal Study of Pregnancy and Childhood (ALSPAC). (28th November 1989); Southmead Health Authority: 49/89 Children of the Nineties -"ALSPAC". (5th April 1990); Frenchay Health Authority: 90/8 Children of the Nineties. (28th June 1990).MAAS: The study was approved by the North West - Greater Manchester East Research Ethics Committee. ERP/94/032 Up to 5 yrs. Allergen avoidance, Primary Prevention, genetics, sRaw age 3 and 5; SOU/00/259 5 year; ERP/95/137 Exposure to pet allergens, atopy, genetics; ERP/97/023 IFWIN, genetics; 03/SM/400 8 year; 06/Q1403/142 10-12 years; 11/NW/0228 13-15 years; 14/NW/1309 18+ years.SEATON: The study was approved by the North of Scotland Research Ethics Committee. REC reference: 13/NS/0108; Protocol number:

2/048/13; Amendment number: AM03.Ashford: The Asthma in Ashford study was reviewed by the Imperial College Research Ethics Committee on 11/11/2014. On 08/01/2015 the Joint Research Compliance Office granted full approval of the study on the basis described in the revised documents. ICREC reference: 14|C2288.IOW: Ethics approval for the IoW cohort was originally given by the Isle of Wight local research ethics committee in 1989 and at each subsequent follow up (1,2 and 4 years) (this is pre "numbers")Age 10 follow up (including DNA and genotyping): Isle of Wight Health Authority Local Research Ethics Committee 18/98. Age 18 Follow up(including DNA and genotyping): Isle of Wight, Portsmouth & South East Hampshire Research Ethics Committee 06/Q1701/34.

In accordance with the Animals (scientific procedures) act 1986, all animal experiments were conducted under the approved UK Home Office Project License No: PPL 70/7643, reviewed by Imperial College's Animal Welfare and Ethical Review body.

## Decision letter and Author response

Decision letter https://doi.org/10.7554/eLife.84315.sa1
Author response https://doi.org/10.7554/eLife.84315.sa2

---

# Additional files

## Supplementary files

- MDAR checklist
- Reporting standard 1. STROBE flowchart.

## Data availability

The informed consent obtained from all included participants does not allow the data to be made freely available through any third party maintained public repository. However, data used for this submission can be made available on request to the corresponding cohort Executive. Researchers will need to submit a research proposal to each cohort Executive Committee. Data access will have a cost, for more details re. ALSPAC contact alspac-data@bristol.ac.uk, for any other cohort contact philip. couch@manchester.ac.uk.The ALSPAC website provides information on how to request and access its data (http://www.bristol.ac.uk/alspac/researchers/access/). For queries regarding access of data from MAAS, IoW, SEATON or Ashford please contact Philip Couch (philip.couch@manchester.ac.uk). All code used to analyse the individual level data and all summary data and code used to plot the figures in our manuscript has been deposited in Dryad.

The following dataset was generated:

| Author(s) | Year | Dataset title | Dataset URL | Database and Identifier |
|---|---|---|---|---|
| Granell R | 2023 | A meta-analysis of genome-wide association studies of childhood wheezing phenotypes identifies ANXA1 as a susceptibility locus for persistent wheezing (GWAS ANXA1) | https://doi.org/10.5061/dryad.3r2280gm3 | Dryad Digital Repository, 10.5061/dryad.3r2280gm3 |

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

## Appendix 1

## Description of cohorts

The STELAR consortium (*Custovic et al., 2015*) brings together five UK population-based birth cohorts as described below. Informed consent was obtained from parents, and study subjects gave their assent/consent when applicable. Data were harmonised and imported into Asthma eLab web-based knowledge management platform to facilitate joint analyses (https://www.asthmaelab.org) (*Custovic et al., 2015*).

### ALSPAC

ALSPAC is a birth cohort study established in 1991 in Avon, UK (*Boyd et al., 2013*; *Fraser et al., 2013*). Pregnant women with expected dates of delivery 1 April 1991 to 31 December 1992 were invited to take part in the study. The initial number of pregnancies enrolled is 14,541. Of these initial pregnancies, there was a total of 14,676 foetuses, resulting in 14,062 live births and 13,988 children who were alive at 1 year of age.

When the oldest children were approximately 7 years of age, an attempt was made to bolster the study with eligible cases who had failed to join originally. As a result, when considering variables collected from the age of 7 onwards (and potentially abstracted from obstetric notes) there are data available for more than the 14,541 pregnancies mentioned above. The number of new pregnancies not in the initial sample (known as Phase I enrolment) that are currently represented on the built files and reflecting enrolment status at the age of 24 is 913 (456, 262, and 195 recruited during Phases II, III, and IV, respectively), resulting in an additional 913 children being enrolled. The phases of enrolment are described in more detail in the cohort profile paper and its update. The total sample size for analyses using any data collected after the age of 7 is therefore 15,454 pregnancies, resulting in 15,589 foetuses. Of these 14,901 were alive at 1 year of age.

Ethical approval: Ethical approval for the study was obtained from the ALSPAC Ethics and Law Committee and the Local Research Ethics Committees. All self-completion questionnaire content is approved by the ALSPAC Ethics and Law Committee. Ethics protocols' numbers: Initial approval Bristol and Weston Health Authority: E1808 Children of the Nineties: Avon Longitudinal Study of Pregnancy and Childhood (ALSPAC) (28 November 1989). Southmead Health Authority: 49/89 Children of the Nineties – 'ALSPAC' (5 April 1990). Frenchay Health Authority: 90/8 Children of the Nineties (28 June 1990).

Informed consent for the use of data collected via questionnaires and clinics was obtained from participants following the recommendations of the ALSPAC Ethics and Law Committee at the time. Data dictionary: The study website contains details of available data through a fully searchable data dictionary: http://www.bristol.ac.uk/alspac/researchers/our-data/.

We are extremely grateful to all the families who took part in this study, the midwives for their help in recruiting them, and the whole ALSPAC team, which includes interviewers, computer and laboratory technicians, clerical workers, research scientists, volunteers, managers, receptionists, and nurses.

### MAAS

MAAS is an unselected birth cohort study established in 1995 in Manchester, UK (*Custovic et al., 2002*). It consists of a mixed urban-rural population within 50 square miles of South Manchester and Cheshire, UK, located within the maternity catchment area of Wythenshawe and Stepping Hill Hospitals. All pregnant women were screened for eligibility at antenatal visits (8–10th week of pregnancy). Of the 1499 couples who met the inclusion criteria (≤10 weeks of pregnancy, maternal age ≥18 years, and questionnaire and skin prick data test available for both parents), 288 declined to take part in the study and 27 were lost to follow-up between recruitment and the birth of a child. A total of 1184 children were born into the study between February 1996 and April 1998. They were followed prospectively for 19 years to date and attended follow-up clinics for assessments, which included lung function measurements, skin prick testing, biological samples (serum, plasma, and urine), and questionnaire data collection. The study was approved by the North West – Greater Manchester East Research Ethics Committee. Ethics protocols' numbers: ERP/94/032 Up to 5 years. Allergen avoidance, primary prevention, genetics, sRaw age 3 and 5; SOU/00/259 5 years; ERP/95/137 Exposure to pet allergens, atopy, genetics; ERP/97/023 IFWIN, genetics; 03/SM/400 8 years; 06/Q1403/142 10–12 years; 11/NW/0228 13–15 years; 14/NW/1309 18+ years.

## SEATON

SEATON is an unselected birth cohort study established in 1997 in Aberdeen, UK, which was designed to explore the relationship between antenatal dietary exposures and asthma outcomes in childhood (*Martindale et al., 2005*). Two-thousand healthy pregnant women attending an antenatal clinic, at median 12 weeks gestation, were recruited. An interviewer administered a questionnaire to the women and atopic status was ascertained by skin prick test (SPT). The cohort included 1924 children born between April 1998 and December 1999. Participants were recruited prenatally and followed up by self-completion questionnaire to 15 years of age using postal questionnaires to record the presence of asthma and allergic diseases. Lung function measurements and SPT to common allergens were performed at 5, 10, and 15 years. The study was approved by the North of Scotland Research Ethics Committee. Ethics protocol REC reference: 13/NS/0108; protocol number: 2/048/13; amendment number: AM03.

## Ashford

The Ashford study is an unselected birth cohort study established in 1991 in Ashford, UK (*Atkinson et al., 1999*). It included 642 children born between 1992 and 1993. Participants were recruited prenatally and followed to age 14 years. Detailed standardised questionnaires were administered at each follow-up to collect information on the natural history of asthma and other allergic diseases. Lung function measurements and SPT was carried out at 5, 8, and 14 years of age. In 2015, the study children aged 20 were sent a self-completion questionnaire, which was returned by 60% of the participants. The Asthma in Ashford study was reviewed by the Imperial College Research Ethics Committee on 11 November 2014. On 8 January 2015 the Joint Research Compliance Office granted full approval of the study on the basis described in the revised documents. ICREC reference: 14|C2288.

## The IOW cohort

IOW is an unselected birth cohort study established in 1989 on the IOW, UK (*Arshad et al., 2018*; *Kurukulaaratchy et al., 2002*; *Kurukulaaratchy et al., 2003*). After the exclusion of adoptions, perinatal deaths, and refusal for follow-up, written informed consent was obtained from parents to enrol 1456 newborns born between 1 January 1989 and 28 February 1990. Follow-up-up assessments were conducted to 26 years of age to prospectively study the development of asthma and allergic diseases. At each follow-up, validated questionnaires were completed by the parents. Additionally, the SPT was performed on 980, 1036, and 853 participants at 4, 10, and 18 years of age to check allergic reactions to common allergens. At 10, 18, and 26 years, spirometry and methacholine challenge tests were performed to diagnose lung problems. Ethics protocols' numbers: Ethics approval for the IoW cohort was originally given by the Isle of Wight Local Research Ethics Committee (now named the National Research Ethics Service, NRES Committee South Central – Southampton B) in 1989 and at each subsequent follow-up (1, 2, and 4 years) (this is pre 'numbers'); age 10 follow-up (including DNA and genotyping): Isle of Wight Health Authority Local Research Ethics Committee 18/98; age 18 follow-up (including DNA and genotyping): Isle of Wight, Portsmouth & South East Hampshire Research Ethics Committee 06/Q1701/34.

## Definition of variables

A list of all variables used in the current study, per cohort, is shown in *Appendix 1—table 1*.

### Demographic, exposures and outcomes

Postal questionnaires were used in ALSPAC and SEATON, while interviewer-administered questionnaires were employed in other cohorts.

Parental history of asthma, eczema, and hay fever was defined based on the responses given to the question 'have you (and/or your partner) ever had asthma/eczema/hay fever'. Maternal and paternal smoking were defined based on the response given to the question 'do you (or does your partner) smoke', administered during pregnancy. Low birth weight was defined as birth weight less than 2500 g based on NHS birth records.

Asthma in MAAS was defined as a case if positive for two of the following criteria: doctor diagnosis of asthma in the past 12 months, current wheeze in the last 12 months, doctor prescription for asthma. Asthma in ALSPAC was defined as a mothers' report of doctor ever diagnosis of asthma.

Current wheeze in MAAS was defined as a questionnaire report to the question 'have you wheezed in the last 12 months' upon attendance at a follow-up clinic. Current wheeze in ALSPAC was defined as a mothers' report to the question 'has your child had any wheezing or whistling in the last 12 months?'.

Asthma medication in ALSPAC was defined as a mothers' report to the question 'has your child taken any asthma medication in the last 12 months?'. Lower respiratory hospital admissions: Data on hospital admissions in MAAS were obtained by manually inspecting the general practice (GP) records for each individual.

Early-life risk factors were divided into four groups according to timing of exposure: maternal and child characteristics (gender, maternal smoking during pregnancy, and maternal history of asthma), perinatal (low birth weight adjusted for gestational age), environmental (pet ownership, smoke exposure after birth), and allergic sensitisation (defined based on positive SPT to cat, HDM, or grass) variables.

## Primary outcome: joint wheeze phenotypes

We used LCA to identify longitudinal trajectories of wheeze (*Oksel et al., 2019a*) based on pooled analysis among 15,941 children with at least two observations on wheezing at five time periods that were approximately shared across all cohorts: infancy (½–1 year); early childhood (2–3 years); pre-school/early school age (4–5 years); middle childhood (8–10 years); and adolescence (14–18 years). Cohort-specific definitions and other variables derived from the questionnaires are provided in *Appendix 1—table 2*.

To control for cohort-specific variation, cohort ID was included in the LCA model as an additional predictor by transforming the five-category variable into a set of four dummy variables and including them as covariates. The largest cohort, ALSPAC, was treated as the non-coded category to which all other cohorts were compared. The expectation maximisation algorithm was used to estimate relevant parameters, with 100,000 iterations and 500 replications.

To assess model fit, we used (1) the Bayesian information criterion (BIC), (2) the Akaike information criterion (AIC), (3) Lo-Mendell-Rubin likelihood ratio test, (4) bootstrapped likelihood ratio, and (4) quality of classification certainty (model entropy). The BIC is an index used in Bayesian statistics to choose among a set of competing models; the model with the lowest BIC is preferred. Using the lowest BIC as a selection criterion, the best fitting model was chosen as the five-class solution with a nominal covariate (BIC:31340). Analyses were carried out using Mplus 8, R (https://www.r-project.org/) and Stata 14 (StataCorp, College Station, TX, USA).

Based on the statistical fit, a five-class solution was selected as the optimal model (*Oksel et al., 2019a*), and the classes (wheeze phenotypes) were labeled as: (1) *never/infrequent wheeze* (52.4%); (2) *early-onset pre-school remitting wheeze* (18.6%), with high prevalence of wheeze during infancy, decreasing to 20% around early-childhood and to less than 10% afterwards; (3) *early-onset middle-childhood remitting wheeze* (9.8%), with early-onset wheeze and peak prevalence in early-childhood (~70%), and diminishing by middle-childhood (<5%); (4) *early-onset persistent wheeze* (10.4%) with 58% wheeze prevalence during infancy, and prevalence between 70–80% thereafter; (5) *late-onset wheeze* (8.8%) with very low prevalence until middle childhood, increasing rapidly to 55% in adolescence. These latent classes were used in the subsequent GWAS.

## Minimising bias and missing data effects

Extracted from reference (*Oksel et al., 2019a*): "One of the advantages of our multicohort approach is that individual studies that might not provide conclusive evidence to make inference about the general population because of cohort specific effects and biases can contribute to revealing a more accurate picture when integrated together. The integration of five cohorts and their pooled analysis enhanced the credibility and generalizability of the phenotyping results to the U.K. population. A further advantage is to minimize the study-specific biases (including cohort specific effects, attrition effects, different recruitment strategies, and geographic factors) affecting the certainty of allocation of individuals to each latent class, while maximizing the benefits of individual cohort studies (e.g., potentially important risk factors and outcomes are captured in some, but not all cohorts)."

"Another strength of pooling cohort data is that a multicohort design allowed us to analyze a large sample with complete data on wheeze from birth to adolescence, thus increasing statistical power to detect less prevalent phenotypes." However, "The optimal solution in the model using

15,941 children (allowing for missing data) remained five classes (see Table E3, Figure E1), and was very similar to that derived from a complete data set." We used results from the larger sample, that is individuals with at least two observations of wheezing, to assign individuals to their most likely wheezing phenotype and used this as our primary outcome in this study.

## Included vs. excluded participants

Related and non-European individuals were excluded as well as those individuals with missing genetic data.

In ALSPAC, 11,176 individuals had data on wheezing phenotypes, of these 6833 were white unrelated and had genetic data. We found more children from mothers who smoked during pregnancy in the excluded sample compared to the included sample; no difference in gender, maternal history of asthma, current wheezing at 8 or 15 years, and small evidence for more asthma ever and current medication at 8 years in the excluded sample (*Appendix 1—table 4*).

In MAAS, 1150 individuals had data on wheezing phenotypes, of these 887 were white unrelated and had genetic data. We found no difference in children from mothers who smoked during pregnancy in the excluded sample compared to the included sample; no difference in gender, maternal history of asthma or current wheezing at both 8 and 16 years. There was small evidence for more asthma ever and current medication at 8 years in the excluded sample (*Appendix 1—table 4*).

In SEATON, 1535 individuals had data on joint wheezing phenotypes, of these 548 were white unrelated and had genetic data. We found evidence for more children from mothers who smoked during pregnancy in the excluded sample compared to the included sample; and more males in the excluded sample. There was no difference in maternal history of asthma or current wheezing, asthma ever or current medication at both 10 and 15 years in the excluded sample compared to the included sample (*Appendix 1—table 4*).

In Ashford 620 individuals had data on joint wheezing phenotypes, of these 348 were white unrelated and had genetic data. We found evidence for more children from mothers who smoked during pregnancy in the excluded sample compared to the included sample; no difference in gender, maternal history of asthma, or asthma ever. There was small evidence for less current wheezing at 8 years, or current medication at 8 years in the excluded sample compared to the included sample (*Appendix 1—table 4*).

In IOW, 1460 individuals had data on joint wheezing phenotypes, of these 952 were white unrelated and had genetic data. We found evidence for more children from mothers who smoked during pregnancy in the excluded sample compared to the included sample; no difference in gender, maternal history of asthma, asthma ever at 10 and 18 years in the excluded sample compared to the included sample. There was small evidence for more children with current wheeze and medication at 8 years in the included sample compared to the included sample (*Appendix 1—table 4*).

**Appendix 1—table 1.** Definition of variables in each of the five Study Team for Early Life Asthma Research (STELAR) birth cohorts.

| Variable | Definition |
| --- | --- |
| *Cohort: ALSPAC* | |
| *Mother – asthma* | Have you ever had asthma? (recruitment) |
| *Mother smoking* | Mother smoked when expecting (recruitment) |
| *Doctor-diagnosed asthma ever* | Has a doctor ever said that your child has asthma? (years 8 and 14) |
| *Current wheezing* | Two questions combined: Occurrence of wheezing and/or wheezing with whistling on the chest in the last 12 months (year ½, 2½, 4¾, 8½, and 14) |
| *Current asthma medication* | Asthma medication in the last 12 months (years 8½ and 14) |
| *Current rhinitis* | Child had problem with sneezing/runny nose without cold/flu in last 12 months (years 7 and 16½) |
| *Current hay fever* | Child had hay fever in last 12 months (years 10½ and 14) |
| *Cohort: MAAS* | |

*Appendix 1—table 1 Continued on next page*

*Appendix 1—table 1 Continued*

| Variable | Definition |
| --- | --- |
| *Mother – asthma* | Has a doctor ever told you that you had asthma? (recruitment) |
| *Mother smoking* | Do you smoke – mother (recruitment) |
| *Doctor-diagnosed asthma ever* | Has your doctor ever told you that your child has or had asthma? (years 8 and 16) |
| *Asthma ever* | Has your child ever suffered from asthma (years 8 and 16) |
| *Current wheezing* | Has your child had wheezing or whistling in the chest in the last 6/12 months (years 1, 3, 5, 8, and 16) |
| *Current asthma medication* | Asthma medication in the last 12 months (years 8 and 16) |
| *Current rhinitis* | Has your child ever had a problem with sneezing, or a runny nose, or a blocked nose when he /she did not have a cold or the flu? (years 8 and 16) |
| *Current hay fever* | Does your child have hay fever now? (years 8 and 16) |
| Cohort: SEATON | |
| *Mother – asthma* | Do you suffer from asthma? (recruitment) |
| *Mother smoking* | Which of the following best describes your smoking status? (recruitment) |
| *Doctor-diagnosed asthma ever* | Has your child ever suffered from asthma? If yes, has this been confirmed by a doctor? (years 10 and 15) |
| *Asthma ever* | Has your child ever suffered from asthma? (year 10); Have you ever suffered from asthma? (year 15) |
| *Current wheezing* | Has your child had wheezing in the chest in the last 12 months (years 1, 2, 5, 10, and 15) |
| *Current asthma medication* | Has your child been prescribed medicines/inhalers for asthma in the last 12 months? (year 10); Have you been prescribed medicines/inhalers for asthma in the last 12 months? (year 15) |
| *Current hay fever* | Has your child suffered from hay fever last 12 months? (years 10 and 15) |
| Cohort: Ashford | |
| *Mother – asthma* | Do you have or have you ever been told you have asthma? (recruitment) |
| *Mother smoking* | Do you smoke cigarettes? (recruitment) |
| *Doctor-diagnosed asthma ever* | Has your doctor ever told you that your child has or had asthma? (years 8 and 14) |
| *Asthma ever* | In the past 12 months has your daughter suffered from asthma? (year 8); Has she/he ever suffered from asthma? (year 14) |
| *Current wheezing* | Which one best describes your child's wheeze in past 12 months? 'Yes' (B:1–6, C:7+), 'No' (A:0) (years 1, 2, 5, 8, and 14) |
| *Current asthma medication* | Over the last 12 months has your daughter taken any of the following treatments (preventer, reliever, nebuliser, steroids) for asthma? (years 8 and 14) |
| *Current rhinitis* | In the last 12 months has your child had a problem with sneezing or a runny or blocked nose? (years 8 and 14) |
| *Current hay fever* | In your opinion does your child have hay fever now? (year 8) Has your child ever had hay fever? (year 14) |
| Cohort: IOW | |
| *Mother – asthma* | Do you or have you suffered from asthma or wheezing (recruitment) |
| *Mother smoking* | Do you smoke in the house? (recruitment) |
| *Doctor-diagnosed asthma ever* | Asthma cared for by hospital specialist/ GP or nurse (years 10, 18, and 26) |
| *Asthma ever* | Child ever had asthma (years 10 and 18) |

*Appendix 1—table 1 Continued on next page*

*Appendix 1—table 1 Continued*

| Variable | Definition |
|---|---|
| *Current wheezing* | Presence of wheeze since previous review (years 1, 2, 4, 10, and 18) |
| *Asthma medication ever* | Child ever had asthma treatment (year 18) combined with asthma treatment questions being asked at years 1, 2, 4, 10, and 18 |
| *Current rhinitis* | In the past 12 months have you had a problem with sneezing, or a runny or blocked nose when you did not have a cold or the flu? (years 10, 18, and 26) |

**Appendix 1—table 2.** The cohort-specific time points and sample size used to ascertain wheeze phenotypes.

| Birth cohort: | IOW | MAAS | SEATON | Ashford | ALSPAC |
|---|---|---|---|---|---|
| *Year of birth* | 1989 | 1995 | 1997 | 1992 | 1991 |
| *Questionnaire* | Interviewer-administered | Interviewer-administered | Postal | Interviewer-administered | Postal |
| *Data collection age (years)* | 1, 2, 4, 10, 18 | 1, 3, 5, 8, 16 | 1, 2, 5, 10, 15 | 1, 2, 5, 8, 14 | ½, 2½, 4¾, 8½, 14 |
| *No. of children with ≥2 observations on wheezing at five selected time points* | 1460 | 1150 | 1535 | 620 | 11,176 |

**Appendix 1—table 3.** Characteristics of the participants in Study Team for Early Life Asthma Research (STELAR) cohorts included in this analysis (restricted to individuals with genetic data).
Numbers are N (%) except for age, where we report mean (SD).

| | ALSPAC | | MAAS | | SEATON | | Ashford | | IOW | |
|---|---|---|---|---|---|---|---|---|---|---|
| | N=6,833 (71.4%) | | N=887 (9.3%) | | N=548 (5.7%) | | N=348 (3.6%) | | N=952 (9.9%) | |
| Males | 3492 (51.1) | | 475 (53.6) | | 260 (47.5) | | 179 (51.4) | | 466 (49.0) | |
| Maternal history of asthma | 748 (11.5) | | 120 (13.5) | | 77 (14.1) | | 49 (14.1) | | 106 (11.2) | |
| Maternal smoking | 1423 (22.1) | | 122 (13.8) | | 107 (19.5) | | 52 (14.9) | | 217 (23.1) | |
| **Wheeze phenotypes** | | | | | | | | | | |
| *Never/infrequent* | 4331 (63.4) | | 506 (57.1) | | 332 (60.6) | | 145 (41.7) | | 573 (60.2) | |
| *Early-onset persistent* | 656 (9.6) | | 133 (15.0) | | 36 (6.6) | | 41 (11.8) | | 77 (8.1) | |
| *Early-onset pre-school remitting* | 1076 (15.8) | | 145 (16.4) | | 117 (21.4) | | 145 (41.7) | | 0 | |
| *Early-onset mid-childhood remitting* | 474 (6.9) | | 48 (5.4) | | 13 (2.4) | | 13 (3.7) | | 55 (5.8) | |
| *Late-onset* | 296 (4.3) | | 55 (6.2) | | 50 (9.1) | | 4 (1.2) | | 247 (26.0) | |

| | 7–8 years | 14–15 years | 8 years | 16 years | 10 years | 15 years | 8 years | 14 years | 10 years | 18 years |
|---|---|---|---|---|---|---|---|---|---|---|
| Age mean (SD) in years | 8.7 (0.3) | 15.4 (0.3) | 7.98 (0.16) | 16.09 (0.62) | 10.15 (0.18) | 15.09 (0.28) | 7.97 (NA) | 13.95 (NA) | 9.98 (0.27) | 17.87 (0.59) |
| Doctor-diagnosed asthma ever* | 1060 (19.7) | 796 (23.2) | 198 (23.9) | 198 (30.0) | 86 (16.0) | 80 (19.5) | 75 (21.6) | 83 (23.9) | 350 (40.9) | 255 (28.6) |
| Asthma ever | NA | NA | 193 (22.8) | 192 (29.5) | 87 (16.2) | 66 (21.9) | 54 (15.6) | 65 (18.7) | 194 (20.9) | 264 (29.3) |
| Current wheeze | 683 (12.5) | 306 (9.0) | 150 (17.6) | 112 (16.9) | 67 (12.4) | 63 (15.5) | 54 (15.6) | 54 (15.5) | 190 (20.4) | 227 (25.1) |
| Current asthma medication | 695 (12.9) | 361 (10.6) | 141 (16.5) | 114 (17.1) | 68 (12.6) | 58 (14.0) | 50 (14.41) | 49 (14.1) | 41 (11.81) | 38 (10.9) |

*DDA ever not available in IOW, we used asthma cared for by hospital specialist/ GP or nurse as proxy.

**Appendix 1—table 4.** Comparison of included vs. excluded participants in the five cohorts at different ages.

| ALSPAC | N | Included | N | Excluded | p-Value |
|---|---|---|---|---|---|
| Males (%) | 6833 | 3492 (51.1) | 4343 | 2269 (52.2) | 0.24 |

*Appendix 1—table 4 Continued on next page*

*Appendix 1—table 4 Continued*

| **ALSPAC** | **N** | **Included** | **N** | **Excluded** | **p-Value** |
|---|---|---|---|---|---|
| Maternal history asthma (%) | 6497 | 748 (11.5) | 4038 | 453 (11.2) | 0.64 |
| Maternal smoking-pregnancy (%) | 6438 | 1423 (22.1) | 4019 | 1167 (29.0) | <0.001 |

| | At 7.5–8.5 years | | | | | At 14–15 years | | | | |
|---|---|---|---|---|---|---|---|---|---|---|
| ALSPAC | N | Included | N | Excluded | p-Value | N | Included | N | Excluded | p-Value |
| Age mean (SD) years | 5139 | 8.7 (0.3) | 1872 | 8.7 (0.3) | <0.001 | 3885 | 15.4 (0.3) | 1237 | 15.5 (0.4) | <0.001 |
| Current wheeze (%) | 5453 | 683 (12.5) | 2579 | 344 (13.3) | 0.308 | 3419 | 306 (9.0) | 1078 | 105 (9.7) | 0.432 |
| Asthma ever (%) | 5377 | 1060 (19.7) | 2605 | 562 (21.6) | 0.053 | 3425 | 796 (23.2) | 1079 | 279 (25.9) | 0.079 |
| Current asthma medication (%) | 5379 | 695 (12.9) | 2529 | 368 (14.6) | 0.047 | 3400 | 361 (10.6) | 1077 | 134 (12.4) | 0.096 |

| **MAAS** | **N** | **Included** | **N** | **Excluded** | **p-Value** |
|---|---|---|---|---|---|
| Males (%) | 887 | 475 (53.6) | 263 | 149 (56.7) | 0.38 |
| Maternal history asthma (%) | 886 | 120 (13.5) | 259 | 45 (17.4) | 0.12 |
| Maternal smoking* (%) | 884 | 122 (13.8) | 260 | 47 (18.1) | 0.09 |

| | At 8 years | | | | | At 16 years | | | | |
|---|---|---|---|---|---|---|---|---|---|---|
| MAAS | N | Included | N | Excluded | p-Value | N | Included | N | Excluded | p-Value |
| Age mean (SD) years | 827 | 7.98 (0.16) | 149 | 8.00 (0.21) | 0.31 | 605 | 16.09 (0.62) | 59 | 15.98 (0.60) | 0.20 |
| Current wheeze (%) | 853 | 150 (17.6) | 172 | 35 (20.4) | 0.39 | 664 | 112 (16.9) | 82 | 15 (18.3) | 0.11 |
| Asthma ever (%) | 845 | 193 (22.8) | 173 | 52 (30.1) | 0.043 | 651 | 192 (29.5) | 79 | 28 (35.4) | 0.28 |
| Current asthma medication (%) | 855 | 141 (16.5) | 173 | 43 (24.9) | 0.009 | 666 | 114 (17.1) | 83 | 14 (16.9) | 0.96 |

| **SEATON** | **N** | **Included** | **N** | **Excluded** | **p-Value** |
|---|---|---|---|---|---|
| Males (%) | 548 | 260 (47.5) | 987 | 525 (53.2) | 0.031 |
| Maternal history asthma (%) | 548 | 77 (14.1) | 985 | 161 (16.4) | 0.24 |
| Maternal smoking* (%) | 548 | 107 (19.5) | 987 | 276 (28.0) | <0.001 |

| | At 10 years | | | | | At 15 years | | | | |
|---|---|---|---|---|---|---|---|---|---|---|
| SEATON | N | Included | N | Excluded | p-Value | N | Included | N | Excluded | p-Value |
| Age mean (SD) years | 548 | 10.15 (0.18) | 987 | 10.23 (0.16) | <0.001 | 545 | 15.09 (0.28) | 916 | 15.11 (0.26) | 0.20 |
| Current wheeze (%) | 541 | 67 (12.4) | 376 | 42 (11.2) | 0.58 | 407 | 63 (15.5) | 310 | 48 (15.5) | 0.99 |
| Asthma ever (%) | 537 | 87 (16.2) | 374 | 53 (14.2) | 0.40 | 409 | 66 (21.9) | 302 | 85 (20.8) | 0.73 |
| Current asthma medication (%) | 542 | 68 (12.6) | 378 | 39 (10.3) | 0.30 | 414 | 58 (14.0) | 309 | 34 (11.0) | 0.23 |

| **Ashford** | **N** | **Included** | **N** | **Excluded** | **p-Value** |
|---|---|---|---|---|---|
| Males (%) | 348 | 179 (51.4) | 272 | 153 (56.3) | 0.23 |
| Maternal history asthma (%) | 348 | 49 (14.1) | 272 | 38 (14.0) | 0.97 |
| Maternal smoking* (%) | 348 | 52 (14.9) | 270 | 61 (22.6) | 0.015 |

| | At 8 years | | | | | At 14 years | | | | |
|---|---|---|---|---|---|---|---|---|---|---|
| Ashford | N | Included | N | Excluded | p-Value | N | Included | N | Excluded | p-Value |
| Age mean (SD) years | 348 | NA | 272 | NA | NA | 348 | NA | 272 | NA | NA |
| Current wheeze (%) | 347 | 54 (15.6) | 246 | 25 (10.2) | 0.06 | 348 | 54 (15.5) | 150 | 18 (12.00) | 0.31 |

*Appendix 1—table 4 Continued on next page*

*Appendix 1—table 4 Continued*

| ALSPAC | N | Included | N | Excluded | p-Value | | | | | |
|---|---|---|---|---|---|---|---|---|---|---|
| Asthma ever (%) | 347 | 54 (15.6) | 246 | 38 (15.5) | 0.97 | 348 | 65 (18.7) | 150 | 25 (16.7) | 0.59 |
| Current asthma medication (%) | 347 | 50 (14.41) | 246 | 22 (8.9) | 0.05 | 348 | 49 (14.1) | 150 | 16 (10.7) | 0.30 |

| IOW | N | Included | N | Excluded | p-Value |
|---|---|---|---|---|---|
| Males (%) | 952 | 466 (49.0) | 508 | 275 (54.1) | 0.06 |
| Maternal history asthma (%) | 946 | 106 (11.2) | 505 | 52 (10.3) | 0.60 |
| Maternal smoking* (%) | 941 | 217 (23.1) | 502 | 147 (29.3) | 0.01 |

| | At 10 years | | | | | At 18 years | | | | |
|---|---|---|---|---|---|---|---|---|---|---|
| IOW | N | Included | N | Excluded | p-Value | N | Included | N | Excluded | p-Value |
| Age mean (SD) years | 932 | 9.98 (0.27) | 426 | 10.04 (0.31) | <0.001 | 914 | 17.87 (0.59) | 389 | 18.14 (0.67) | <0.001 |
| Current wheeze (%) | 932 | 190 (20.4) | 426 | 69 (16.2) | 0.07 | 903 | 227 (25.1) | 377 | 58 (15.4) | <0.002 |
| Asthma ever (%) | 930 | 194 (20.9) | 425 | 80 (18.8) | 0.39 | 900 | 264 (29.3) | 385 | 108 (28.1) | 0.64 |
| Current asthma medication (%) | 347 | 41 (11.81) | 246 | 15 (6.10) | 0.02 | 348 | 38 (10.9) | 150 | 13 (8.7) | 0.45 |

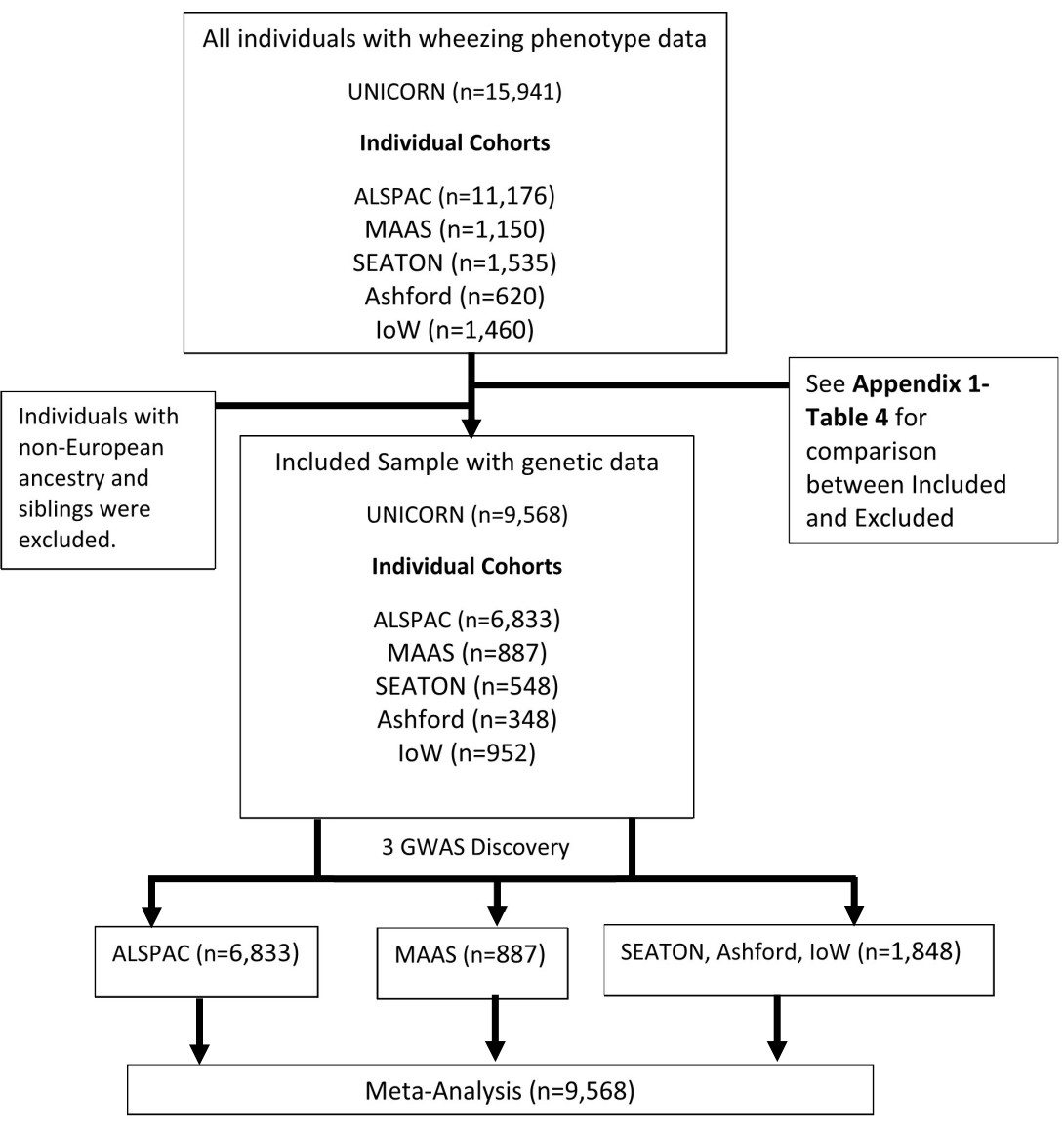

**Appendix 1—figure 1.** Flowchart of individuals included in the final meta-analysis.

## Appendix 2

### Genotyping and imputation

#### ALSPAC

Participants were genotyped using the Illumina HumanHap550 quad genome-wide SNP genotyping platform (Illumina Inc, San Diego, CA, USA) by the Wellcome Trust Sanger Institute (WTSI; Cambridge, UK) and the Laboratory Corporation of America (LCA, Burlington, NC, USA), using support from 23andMe. Haplotypes were estimated using ShapeIT (v2.r644) which uses relationship information to improve phasing accuracy. The phased haplotypes were then imputed to the Haplotype Reference Consortium (HRCr1.1, 2016) panel (*Loh et al., 2016*) of approximately 31,000 phased whole genomes. The HRC panel was phased using ShapeIt v2, and the imputation was performed using the Michigan imputation server.

#### MAAS

In MAAS, we used the Illumina 610 quad genome-wide SNP genotyping platform (Illumina Inc, San Diego, CA, USA). Prior to imputation samples were excluded on the basis of gender mismatches; minimal or excessive heterozygosity, genotyping call rates of <97%. SNPS were excluded if they had call rates of <95%, minor allele frequencies of <0.5%, and HWE $p<3 \times 10^{-8}$. Prior to imputation each chromosome was pre-phased using EAGLE2 (v2.0.5) (*Loh et al., 2016*) as recommended by the Sanger imputation server (*McCarthy et al., 2016*). We then imputed with PBWT (*Durbin, 2014*) with the Haplotype Reference Consortium (release 1.1) of 32,470 reference genomes (*McCarthy et al., 2016*) using the Sanger Imputation Server.

#### IOW, SEATON, and Ashford

IOW, SEATON, and Ashford were genotyped using the illumina Infinium Omni2.5–8 v1.3 BeadChip genotyping platform (Illumina Inc, San Diego, CA, USA). Genotype QC and imputation was carried out as described for MAAS.

#### Exclusions

Individuals were excluded on the basis of gender mismatches; minimal or excessive heterozygosity; disproportionate levels of individual missingness (>3%), insufficient sample replication (IBD <0.8), or evidence of cryptic relatedness (IBD >0.1). Following imputation, SNPs with a minor allele frequency of <1%, a call rate of <95%, evidence for violations of Hardy-Weinberg equilibrium (p<5E-7), or imputation quality measure (MaCH-Rsq or IMPUTE-info score)<0.40 were excluded. All individuals with non-European ancestry and siblings were removed.

### GWAS meta-analysis

GWASs of the joint wheezing phenotypes were performed independently in ALSPAC, MAAS, and the combined IOW-SEATON-Ashford (combined as they were genotyped on the same platform, at the same time, and quality-controlled and imputed together). All genetic data were imputed to a new Haplotype Reference Consortium panel. This comprises around 31,000 sequenced individuals (mostly European), so the coverage of European haplotypes is much greater than in other panels. As a consequence, we expect to improve imputation accuracy, particularly at lower frequencies.

We used SNPTEST v2.5.2 (*Marchini and Howie, 2010*) with a frequentist additive multinomial regression model (-method newml, never/infrequent wheeze as the reference) to investigate the association between SNPs and wheezing phenotypes. No covariates were included in the model and only individuals of European descent were included in this analysis. A meta-analysis of the three GWASs, including 5887 controls and 943 cases for early-onset persistent, 1482 cases for early-onset remitting, 603 cases for mid-childhood onset remitting, and 652 cases for late-onset wheeze, was performed using METAL (*Willer et al., 2010*) with a total of 8,057,852 SNPs present. We used the option SCHEME STDERR in METAL to implement an effect size-based method weighted by each study-specific standard error in a fixed-effects model. We performed clumping to keep only one representative SNP per LD block and used LZPs to short-list independent SNPs for further annotation.

## LD clumping, pre-selection, and gene annotation

LD clumping was performed for all SNPs with p-value $<10^{-5}$ for at least one wheezing phenotype. In order to avoid redundancy between SNPs and to ensure associations are independent, we used significance thresholds of 0.05 for index and clumped SNPs (`--clump-p1` 0.05, `--clump-p2` 0.05), LD threshold of 0.80 (`--clump-r2` 0.80) and physical distance threshold of 250 kb (`--clump-kb` 250). European 1000 Genome data were used to infer LD structure.

LZPs (http://locuszoom.org/) (*Pruim et al., 2010*) were used for close inspection of all independent signals. Loci showing a peak with different colour dots (possibly indicating more than one causal variant) were short-listed for further annotation. SNPnexus database (https://www.snp-nexus.org/v4/) (*Dayem Ullah et al., 2018*) was used to annotate the overlapping, upstream and downstream genes; the GWAS Catalog (by SNP and then gene) (https://www.ebi.ac.uk/gwas/search), GeneCards (https://www.genecards.org/) (*Stelzer et al., 2016*), database, and phenoscanner (http://www.phenoscanner.medschl.cam.ac.uk/) were used to further explore previously associated relevant phenotypes and gene function. Lead SNPs were looked in https://www.regulomedb.org/ to assess potential functionality.

## Genetic control

The genomic inflation factor ($\lambda$) was calculated using the scipy.stats.chi2 module in Python. The chi-squared test statistics from the meta-analysis p-values were first obtained. Then, the observed median chi-squared statistic from the calculated chi-squared test statistics were calculated. Finally, the genomic inflation factor ($\lambda$) was derived by dividing the observed median chi-squared statistic by the expected median chi-squared statistic.

## Heterogeneity scatter plots

Heterogeneity scatter plots were based on filtering signals for each pair of wheezing phenotypes. For example, for group1=persistent and group2=early-onset mid-childhood remitting wheezing.

If group1 has a p-value $<10^{-5}$ and group2 has a p-value $>0.05$, and group1 has a negative effect size (beta) while the lower bound of group2's effect size (beta - CI) is greater than group1's effect size, then we classified the result as group1 specific. If group1 has a p-value $<10^{-5}$ and group2 has a p-value $>0.05$, and group1 has a positive effect size (beta) while the upper bound of group2's effect size (beta+CI) is less than group1's effect size, then we classified the result as group1 specific.

If group2 has a p-value $<10^{-5}$ and group1 has a p-value $>0.05$, and group2 has a negative effect size (beta) while the lower bound of group1's effect size (beta - CI) is greater than group2's effect size, then we classified the result as group2 specific. If group2 has a p-value $<10^{-5}$ and group1 has a p-value $>0.05$, and group2 has a positive effect size (beta) while the upper bound of group1's effect size (beta+CI) is less than group2's effect size, then we classified the result as group2 specific.

If both group1 and group2 have p-values $<10^{-5}$, and their effect sizes (betas) have the same sign (i.e., both positive or both negative), then we classified the result as 'Same direction'.

If both group1 and group2 have p-values $<10^{-5}$, and their effect sizes (betas) have opposite signs (i.e., one positive and one negative), then we classified the result as 'Opposite direction'.

## Gene expression in whole blood and lung tissues

The top independent SNPs associated with each of the wheeze phenotypes were assessed for their association with cis- and trans-acting gene expression (mRNA) in whole blood and lung tissues. We identified potential eQTL signals using Genotype-Tissue Expression database (https://gtexportal.org) using the European reference panel.

**Appendix 2—table 1.** List of 134 independent single nucleotide polymorphisms (SNPs) identified after clumping and associated with at least one wheezing phenotype (p$<10^{-5}$).

| CHR | SNP | BP | Short-listed after inspection of locus zoom plot |
|---|---|---|---|
| Persistent wheezing | | | |
| 1 | rs4620530 | 240063821 | Yes |
| 2 | rs13398488 | 7142199 | Yes |

*Appendix 2—table 1 Continued on next page*

Appendix 2—table 1 Continued

| CHR | SNP | BP | Short-listed after inspection of locus zoom plot |
|---|---|---|---|
| 2 | rs77062323 | 53049017 | No |
| 2 | rs6543291 | 106011626 | Yes |
| 3 | rs77655717 | 128737320 | Yes |
| 4 | rs77822621 | 1008212 | Yes |
| 4 | rs7680608 | 1050437 | Yes |
| 4 | rs115228498 | 142969757 | Yes |
| 4 | rs145937716 | 143192224 | No |
| 5 | rs116494115 | 7736317 | Yes |
| 5 | rs78701483 | 95680422 | No |
| 6 | rs138099941 | 7654240 | No |
| 6 | rs9346404 | 71606613 | No |
| 6 | rs143979498 | 151040328 | No |
| 7 | rs76871421 | 105676144 | Yes |
| 8 | rs59670576 | 128555771 | No |
| 9 | rs116933120 | 27458652 | No |
| 9 | rs75260654 | 75788108 | Yes |
| 9 | rs116849664 | 75820902 | Yes |
| 9 | rs143481506 | 139515723 | No |
| 10 | rs7088157 | 100038964 | Yes |
| 11 | rs112474574 | 5885773 | Yes |
| 11 | rs116861530 | 116962661 | Yes |
| 13 | rs7982350 | 73106322 | No |
| 13 | rs17461573 | 106711373 | No |
| 14 | rs1105683 | 56213787 | Yes |
| 15 | rs2202714 | 49811991 | Yes |
| 15 | rs117540214 | 84338642 | Yes |
| 17 | rs17676191 | 37949924 | Yes |
| 17 | rs79026872 | 37965932 | Yes |
| 17 | rs4795400 | 38067020 | Yes |
| 17 | rs1031460 | 38072247 | Yes |
| 17 | rs56199421 | 38090808 | Yes |
| 17 | rs4795406 | 38100134 | Yes |
| 17 | rs72832972 | 38110575 | Yes |
| 17 | rs4794821 | 38124203 | Yes |
| 17 | rs59843584 | 38124892 | Yes |
| 18 | rs111812993 | 30353181 | No |
| 19 | rs4804311 | 8615589 | Yes |
| 19 | rs2013694 | 8616392 | Yes |

Appendix 2—table 1 Continued on next page

Appendix 2—table 1 Continued

| CHR | SNP | BP | Short-listed after inspection of locus zoom plot |
| --- | --- | --- | --- |
| 19 | rs73501545 | 8620823 | Yes |
| 19 | rs111644945 | 8625081 | Yes |
| 22 | rs5994170 | 17615213 | Yes |
| 22 | rs34902370 | 17632194 | Yes |
| Early-onset remitting wheezing | | | |
| 1 | rs12730098 | 212427488 | Yes |
| 1 | rs75639566 | 233019116 | No |
| 2 | rs2880066 | 17107219 | Yes |
| 2 | rs10180268 | 17126699 | Yes |
| 3 | rs115031796 | 86691640 | No |
| 3 | rs3861377 | 173317378 | Yes |
| 3 | rs10513743 | 176022304 | Yes |
| 5 | rs10075253 | 75548246 | Yes |
| 5 | rs12520884 | 84406634 | No |
| 6 | rs117477297 | 92565052 | No |
| 6 | rs2453395 | 166286532 | Yes |
| 7 | rs56027869 | 50072919 | No |
| 7 | rs4730561 | 78531705 | Yes |
| 7 | rs73144976 | 78586112 | Yes |
| 7 | rs67259321 | 78686582 | Yes |
| 7 | rs146771277 | 154438861 | No |
| 9 | rs10758259 | 34392908 | Yes |
| 11 | rs7128994 | 71242209 | No |
| 11 | rs72994149 | 106837223 | Yes |
| 12 | rs117367256 | 93508478 | No |
| 13 | rs2872948 | 57442480 | Yes |
| 13 | rs73527654 | 57447994 | Yes |
| 13 | rs2151504 | 82291577 | No |
| 15 | rs116966886 | 47043587 | Yes |
| 15 | rs117565527 | 47342882 | Yes |
| Mid-childhood onset remitting wheezing | | | |
| 1 | rs35725789 | 159207367 | Yes |
| 1 | rs146141555 | 159227423 | Yes |
| 1 | rs146575092 | 159374228 | Yes |
| 1 | rs140877050 | 220848829 | No |
| 1 | rs72745905 | 223451086 | No |
| 2 | rs7595553 | 36127878 | Yes |
| 2 | rs145007503 | 50688324 | No |

Appendix 2—table 1 Continued on next page

*Appendix 2—table 1 Continued*

| CHR | SNP | BP | Short-listed after inspection of locus zoom plot |
|-----|-----|-----|-----|
| 2 | rs6546068 | 64583398 | No |
| 2 | rs17387431 | 206651315 | No |
| 2 | rs144791928 | 236963432 | No |
| 3 | rs34315999 | 8969653 | Yes |
| 3 | rs115245770 | 99209128 | No |
| 3 | rs146961758 | 194285978 | Yes |
| 4 | rs138794367 | 103859545 | Yes |
| 5 | rs115719402 | 77538102 | Yes |
| 6 | rs76026399 | 47531792 | No |
| 7 | rs73172838 | 154842348 | No |
| 8 | rs112631708 | 134500083 | No |
| 9 | rs72752356 | 98094970 | No |
| 13 | rs113195384 | 46333770 | No |
| 13 | rs9602218 | 84139813 | Yes |
| 13 | rs61960366 | 84144202 | Yes |
| 13 | rs74589927 | 84208697 | Yes |
| 13 | rs2210726 | 84492936 | Yes |
| 13 | rs4390476 | 84598570 | Yes |
| 14 | rs117443464 | 73460284 | Yes |
| 16 | rs72820814 | 81916262 | No |
| 17 | rs190526697 | 12274299 | No |
| 18 | rs75286534 | 26206826 | No |
| 18 | rs138888086 | 63591085 | No |
| 18 | rs76551535 | 71879807 | No |
| 19 | rs77496444 | 19192132 | No |
| 20 | rs6077514 | 9302948 | Yes |
| Late-onset wheezing | | | |
| 1 | rs9439669 | 18859049 | Yes |
| 1 | rs2051039 | 57067560 | Yes |
| 1 | rs72673642 | 80727443 | Yes |
| 2 | rs147557117 | 19778063 | No |
| 2 | rs140983998 | 111402871 | Yes |
| 2 | rs117617447 | 123387601 | No |
| 2 | rs13025116 | 127505482 | No |
| 2 | rs148008098 | 128633620 | Yes |
| 3 | rs4072729 | 24780393 | Yes |
| 3 | rs143960666 | 31227943 | No |
| 3 | rs4677102 | 72193991 | No |

*Appendix 2—table 1 Continued on next page*

*Appendix 2—table 1 Continued*

| CHR | SNP | BP | Short-listed after inspection of locus zoom plot |
|---|---|---|---|
| 3 | rs145629570 | 113422516 | Yes |
| 3 | rs113643470 | 141728174 | Yes |
| 4 | rs17472015 | 48467594 | Yes |
| 7 | rs117660982 | 149438923 | Yes |
| 7 | rs118027705 | 150456728 | Yes |
| 7 | rs139489493 | 150481499 | Yes |
| 7 | rs144271668 | 157934780 | Yes |
| 8 | rs990182 | 89976447 | Yes |
| 9 | rs79110962 | 14432953 | Yes |
| 10 | rs9325460 | 82492323 | No |
| 10 | rs7896106 | 91196402 | Yes |
| 10 | rs115465993 | 109372900 | No |
| 11 | rs16935643 | 41395746 | No |
| 11 | rs141958628 | 119083284 | Yes |
| 14 | rs113363660 | 69410278 | No |
| 15 | rs139134265 | 44923960 | Yes |
| 15 | rs143862030 | 84922146 | Yes |
| 16 | rs113390367 | 1118849 | Yes |
| 16 | rs4788025 | 28003221 | Yes |
| 18 | rs72918264 | 51009510 | No |
| 22 | rs133498 | 48913809 | Yes |

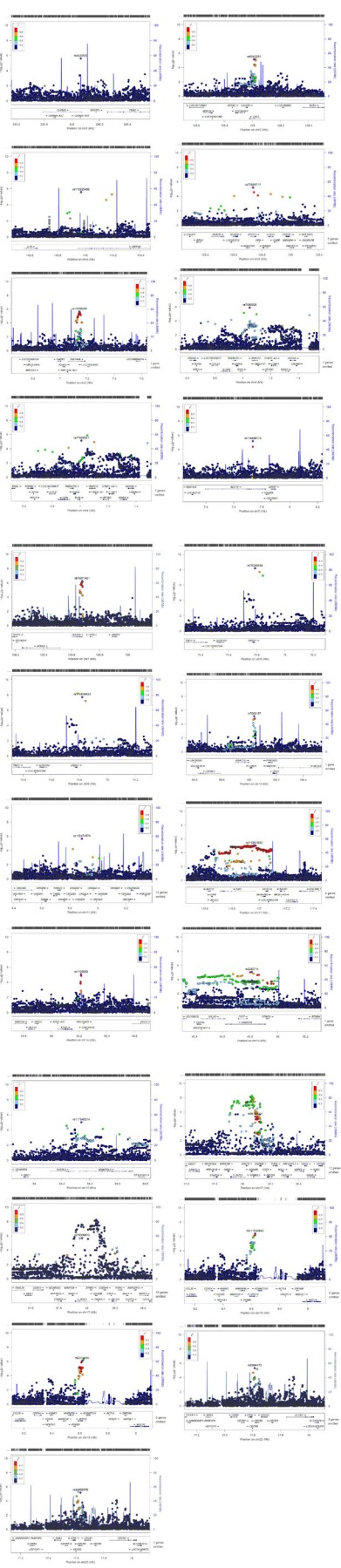

**Appendix 2—figure 1.** Zoom locus plots for short-listed independent top hits for persistent wheezing.

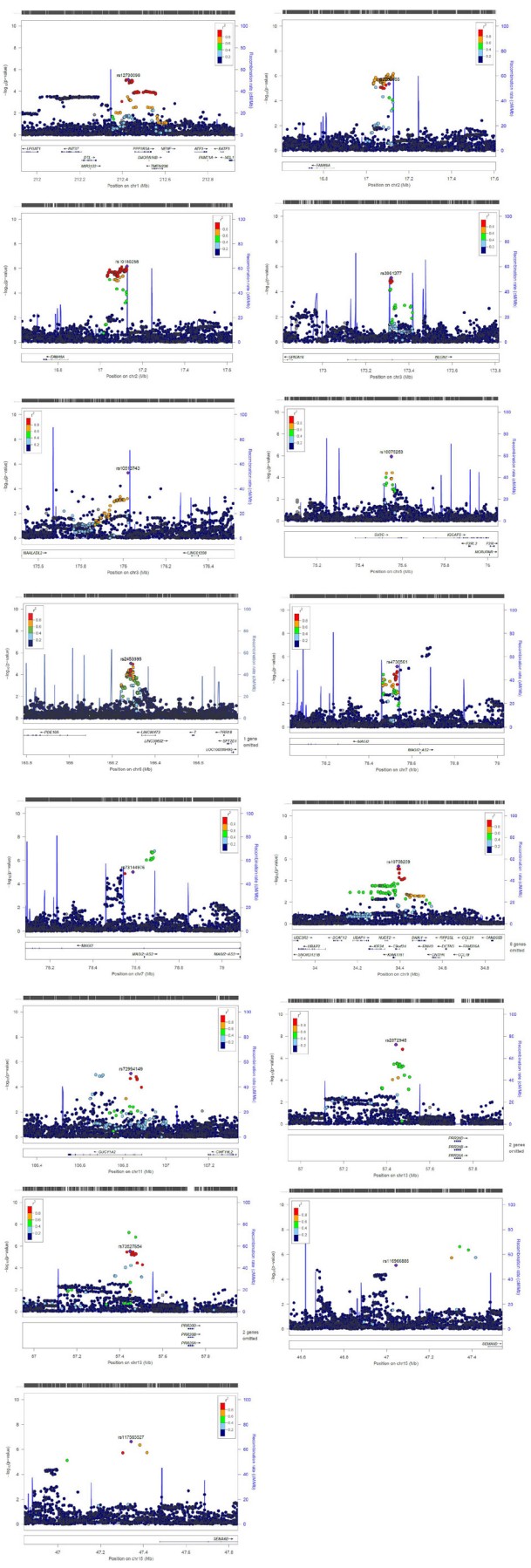

**Appendix 2—figure 2.** Zoom locus plots for short-listed independent top hits for early-onset pre-school remitting wheezing.

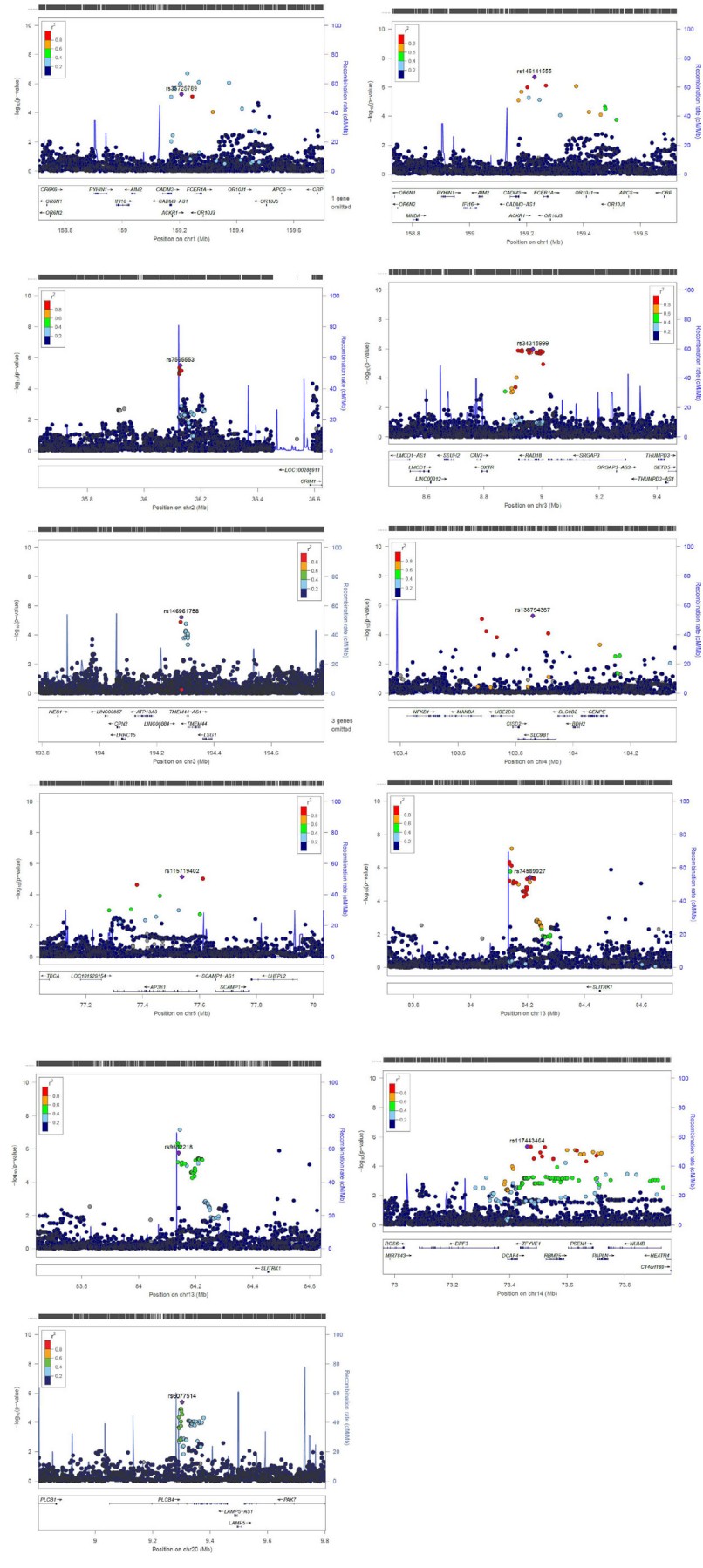

**Appendix 2—figure 3.** Zoom locus plots for short-listed independent top hits for early-onset mid-childhood remitting wheezing.

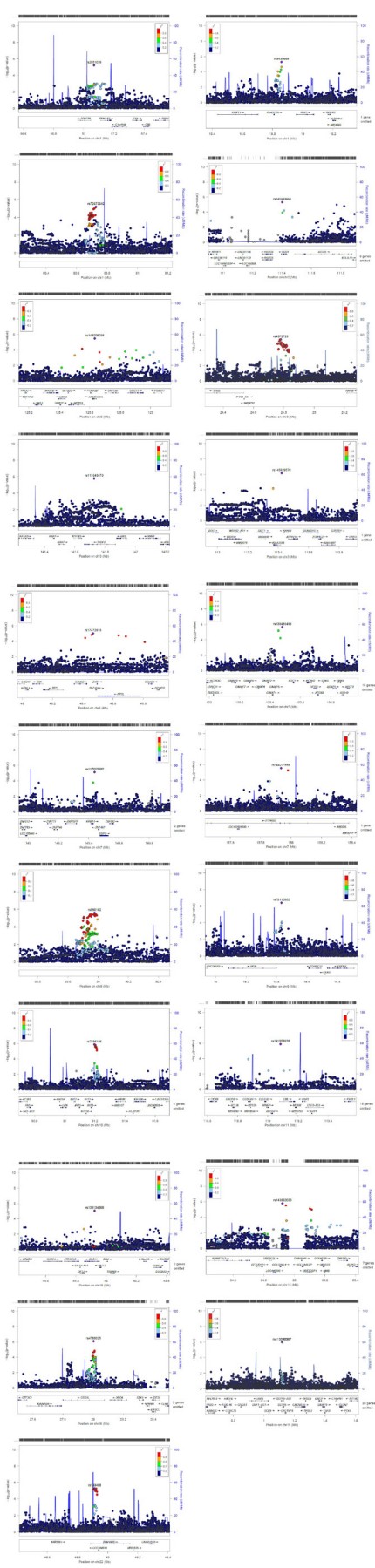

**Appendix 2—figure 4.** Zoom locus plots for short-listed independent top hits for late-onset wheezing.

## Appendix 3

### Post-GWAS: rs75260654 (*ANXA1*)

#### Annotation and distribution

Information including chromosome, strand, clinical significance was retrieved from ENSEMBL using the R package biomaRt (***Durinck et al., 2005***; ***Durinck et al., 2009***). The effects of rs75260654 on genomic features were predicted by querying the Ensembl VEP (***McLaren et al., 2016***) web tool.

rs75260654 distribution in the GRCh38.p13 build of the human genome across African, Asian, and European populations of the 1000 Genomes Project Phase 3 were accessed by querying the Ensembl (https://www.ensembl.org) web browser on 24 May 2021.

#### Promoter Capture

The Hi-C libraries were prepared from CD4+ T cells isolated from seven healthy individuals (two libraries per individual) from the MAAS cohort using the Arima-HiC kit (Arima Genomics). PCHi-C libraries were generated by capturing the restriction fragments (RF) overlapping the TSS of 18,775 protein-coding genes using the Agilent SureSelectXT HS Target Enrichment System according to the manufacturer's protocols. The final design included 305,419 probes covering 13.476 Mb and 18,630 protein-coding genes. The RF overlapping the TSS (±1 RF, 3 RF per promoter) were captured with custom-designed biotinylated RNA baits. Libraries were sequenced to ~300 M 2×150 bp reads each (~600 M reads/individual). The 3'-end of the reads was quality trimmed with Sickle. The sequencing data were processed with the HiCUP pipeline to map the sequencing reads and eliminate experimental artefacts and PCR duplicates (***Wingett et al., 2015***). The BAM files from technical replicates were merged. Promoter interactions were called using the CHiCAGO pipeline (***Cairns et al., 2016***), which calls statistically significant interactions in PCHi-C data while accounting for noise and PCHi-C-specific bias. A CHiCAGO score >5 (soft-thresholded -log weighted p-value) was considered significant. To gain information from all the available data, the BAM files from all seven individuals were supplied as biological replicates in the analysis with CHiCAGO. Moreover, to increase power, RF were binned as follows: 10 consecutive RF that were not covered by the baits were binned together; the 3 baited fragments for each promoter were binned with 1 RF upstream and 1 downstream, totalling 5 fragments per promoter. If the bins for two consecutive promoters overlapped, these were binned together into a single larger bin. Publicly available ENCODE ATAC-seq (A549 cell line) and ChIP-seq (A549 cell line ENCFF900GVO) and POLR2A ChIP-seq (A549 cell line, ENCFF737ZKN) data and 18-state ChromHMM from the EpiMap Project (BSS00007) (***Adsera et al., 2019***) for A549 cell line were downloaded. The PCHi-C interactions of interest and their overlap with ATAC-seq and ChIP-seq peaks, and putative enhancers from the 18-state ChromHMM model were visualised using the Sushi R package.

#### eQTL catalogue lookup

We queried the eQTL catalogue (https://www.ebi.ac.uk/eqtl/; accessed 6 May 2021) using tabix-0.2.6 to assess if *rs75260654*, *rs116849664,* or *rs78320984* are eQTLs in studies that utilised the following cell types: lung, T cells, blood, monocytes, neutrophils, NK cells, fibroblasts, B cells, CD4+ T cells, CD8+ T cells, Th17 cells, Th1 cells, Th2 cells, Treg naive, Treg memory, CD16+ monocytes, cultured fibroblasts, EBV-transformed lymphocytes. We defined nominal significance as $p \leq 0.05$.

#### Variant effect

Variant effect on tissue-specific gene expression, which is based on GTEx eQTL, was retrieved on May 24 from eQTL Ensembl database (https://www.ensembl.org/) and eQTLGene Consortium (https://www.eqtlgen.org/cis-eqtls.html). Using downloaded correlation of variant on tissue-specific gene expression from Ensembl, the relative effect of T allele on the *ANXA1* expression across 86 tissue types was presented in scatter plot using R version 3.6.1 (***Ihaka and Gentleman, 1996***). To get information on the functional role of *ANXA1*, the top 30 interacting proteins and enrichment were retrieved from STRING database (***Szklarczyk et al., 2019***) into cystoscape for visualisation (***Shannon et al., 2003***).

# Appendix 4

## Results in context of literature

Previously relevant associated traits for each region/gene now associated with different wheezing phenotypes are presented in *Appendix 5—tables 1–4*.

## Persistent wheeze

We identified two GWAS-significant loci: 17q21, $p<5.5 \times 10^{-9}$, and a novel locus on 9q21.13 (*ANXA1*), $p<6.7 \times 10^{-9}$. The remaining 17 loci ($4.0 \times 10^{-7} \leq$ p-values $\leq 9.8 \times 10^{-6}$) included regions previously associated with childhood asthma (1q43, 4p16.3, 4q31.21, 5p15.31, 7q22.3, 17q12), asthma and rhinitis (2p25.1), eosinophil count (3q21.3, 10q24.2, 11q23.3, 22q11.1), bronchial hyper-responsiveness (2q12.2), lung function (1q43, 3q21.3, 5q13.3, 15q25.2, 19p13.2), triglycerides measurement and/or glucose metabolism (11q23.3, 19p13.2, and 22q11.1), severe asthma (14q22.1), and severe asthma and insulin resistance (11p15.4). See *Appendix 5—table 1*.

## Early-onset pre-school remitting wheeze

Among the regions associated with early-onset pre-school remitting wheeze, we identified loci previously associated with smoking (3q26.31, 7q21.11, and 15q21.1), waist circumference and obesity (1q32.3), asthma and/or BMI (5q13.3, 6q27, 7q21.11), allergic disease and atopy (7q21.11), and airway repair (2p24.2 and 9p13.3). See *Appendix 5—table 2*.

## Early-onset mid-childhood remitting wheeze

Loci associated with this phenotype were previously associated with neutrophil counts (1q23.2, 3q29, 20p12.3-p12.2), eosinophil counts and allergic rhinitis (4q24), pollution and DNA methylation (2p22.3), atopy (3p25.3), food allergy (13q31.1), and BMI (3q29, 5q14.1). See *Appendix 5—table 3*.

## Late-onset wheeze

Regions associated with late-onset wheeze were previously associated with adult-onset non-allergic asthma (3q13.2), asthma/allergic disease and allergy/atopic sensitisation (3q23, 7q36.1), asthma and/or allergy in adolescence (9p22.3, 16p12.1), late-onset asthma and obesity (2q13), lung function or body height (2q14.3, 3p24.2, 3q13.2, 15q25.2), lymphocyte count and asthma susceptibility (1p32.2), obesity and/or metabolic syndrome/dysfunction (1p36.13 and 22q13.32), eczema (7q36.3), insulin resistance (10q23.31), type 1 diabetes (8q21.3), alcohol drinking (15q15.3-q21.1), and sex hormone-binding globulin levels (11q23.3). See *Appendix 5—table 4*.

**Appendix 4—table 1.** References to previous relevant associated traits for each region/gene identified in early-onset persistent wheezing.

**Early-onset persistent wheezing**

| Gene(s) | Locus | Previous associated trait | Reference or source |
|---|---|---|---|
| *CHRM3* | 1q43 | **FEV₁, FEV₁/FVC, asthma** – high priority drug target | Patel, K.R. et al. Targeting acetylcholine receptor M3 prevents the progression of airway hyperreactivity in a mouse model of childhood asthma. *FASEB J* **31**, 4335–4346 (2017). |
| *RNF144A* | 2p25.1 | **Asthma**, allergy, childhood onset **asthma, allergic rhinitis** | Schoettler, N. et al. Advances in asthma and allergic disease genetics: Is bigger always better? *J Allergy Clin Immunol* **144**, 1495–1506 (2019). |
| *FHL2* | 2q12.2 | **Bronchial hyper-responsiveness**, airway inflammation, novel gene associated with **asthma** severity in human | Kurakula, K. et al. Deficiency of FHL2 attenuates airway inflammation in mice and genetic variation associates with human bronchial hyper-responsiveness. *Allergy* **70**, 1531–44 (2015). |
| *RAB7A* | 3q21.3 | **Eosinophil count** | GeneCards |

*Appendix 4—table 1 Continued on next page*

*Appendix 4—table 1 Continued*

**Early-onset persistent wheezing**

| Gene(s) | Locus | Previous associated trait | Reference or source |
|---|---|---|---|
| *RAB43* | 3q21.3 | Response to **bronchodilator**, FEV$_1$/FEC ratio | GWAS Catalog |
| *RNF212, IDUA, DGKQ, SLC26A1* | 4p16.3 | **Asthma** | Gautam, Y. et al. Comprehensive functional annotation of susceptibility variants associated with asthma. *Hum Genet* **139**, 1037–1053 (2020). |
| *INPP4B* | 4q31.21 | Atopic **asthma** | Sharma, M. et al. A genetic variation in inositol polyphosphate 4 phosphatase a enhances susceptibility to asthma. *Am J Respir Crit Care Med* **177**, 712–9 (2008). |
| *ADCY2* | 5p15.31 | **Asthma**×air pollution, childhood **asthma** | Gref, A. et al. Genome-Wide Interaction Analysis of Air Pollution Exposure and Childhood Asthma with Functional Follow-up. *Am J Respir Crit Care Med* **195**, 1373–1383 (2017). |
| *CDHR3* | 7q22.3 | Childhood **asthma** | Everman, J.L. et al. Functional genomics of CDHR3 confirms its role in HRV-C infection and childhood asthma exacerbations. *J Allergy Clin Immunol* **144**, 962–971 (2019). |
| *ANXA1* | 9q21.13 | FEV$_1$/FVC, response to **bronchodilators** in smokers | Lutz, S.M. et al. A genome-wide association study identifies risk loci for spirometric measures among smokers of European and African ancestry. *BMC Genet* **16**, 138 (2015). |
| *ANXA1* | 9q21.13 | Anti-inflammatory properties, strongly expressed in **bronchial mast cells** | Vieira Braga FA et al. A cellular census of human lungs identifies novel cell states in health and in asthma. (2019). |
| *ANXA1* | 9q21.13 | Potentially involved in epithelial **airway repair** | Leoni, G. et al. Annexin A1-containing extracellular vesicles and polymeric nanoparticles promote epithelial wound repair. *J Clin Invest* **125**, 1215–27 (2015). |
| *R3HCC1L* | 10q24.2 | Atopic **eczema**, psoriasis | GWAS Catalog |
| *R3HCC1L* | 10q24.2 | **Eosinophil count**, BMI | GeneCards |
| *TRIM5, TRIM6, TRIM22* | 11p15.4 | Severe **asthma** and insulin resistance | Kimura, T. et al. Precision autophagy directed by receptor regulators - emerging examples within the TRIM family. *J Cell Sci* **129**, 881–91 (2016). |
| *SIK3* | 11q23.3 | Triglycerides, glucose metabolism, **eosinophil count** | Sun, Z. et al. The potent roles of salt-inducible kinases (SIKs) in metabolic homeostasis and tumorigenesis. Sig Transduct Target Ther **5** (2020). |
| *KTN1* | 14q22.1 | Severe **asthma** | Bigler, J. et al. A Severe Asthma Disease Signature from Gene Expression Profiling of Peripheral Blood from U-BIOPRED Cohorts. Am J Respir Crit Care Med **195**, 1311–1320 (2017). |
| *FAM227B* | 5q13.3 | rs35251997 and FEV$_1$, FEV$_1$/FVC | Shrine, N. et al. New genetic signals for lung function highlight pathways and chronic obstructive pulmonary disease associations across multiple ancestries. *Nat Genet* **51**, 481–493 (2019). |
| *ADAMTSL3* | 15q25.2 | **FEV1/FVC** | Sakornsakolpat, P. et al. Genetic landscape of chronic obstructive pulmonary disease identifies heterogeneous cell-type and phenotype associations. *Nat Genet* **51**, 494–505 (2019). |
| *IKZF3, GSDMB, LRRC3C, GSDMA* | 17q12 | Early-onset **asthma**, **persistent wheezing** (chr17q12-q21) | Granell R et al. Examination of the relationship between variation at 17q21 and childhood wheeze phenotypes. J Allergy Clin Immunol. 2013 Mar;131(3):685–94. |

*Appendix 4—table 1 Continued on next page*

*Appendix 4—table 1 Continued*

**Early-onset persistent wheezing**

| Gene(s) | Locus | Previous associated trait | Reference or source |
|---|---|---|---|
| MARCH2, HNRNPM, MYO1F | 19p13.2 | Triglycerides, HDL cholesterol, metabolic syndrome | Sajuthi, S.P. et al. Genetic regulation of adipose tissue transcript expression is involved in modulating serum triglyceride and HDL-cholesterol. *Gene* **632**, 50–58 (2017). |
| MYO1F | 19p13.2 | **FEV$_1$ and FVC** | GeneCards |
| CECR5 | 22q11.1 | Triglycerides, **eosinophil count**, and body height | Liu, D.J. et al. Exome-wide association study of plasma lipids in >300,000 individuals. *Nat Genet* **49**, 1758–1766 (2017). |

**Appendix 4—table 2.** References to previous relevant associated traits for each region/gene identified in early-onset pre-school remitting wheezing.

**Early-onset pre-school remitting wheezing**

| Gene(s) | Locus | Previous associated trait | Reference or source |
|---|---|---|---|
| PPP2R5A | 1q32.3 | **Waist circumference and obesity** | Kim, H.J. et al. Combined linkage and association analyses identify a novel locus for obesity near PROX1 in Asians. *Obesity (Silver Spring)* **21**, 2405–12 (2013). |
| FAM49A or CYRIA | 2p24.2 | **Airway repair** in **non-atopic asthma** | Hoang, T.T. et al. Epigenome-wide association study of DNA methylation and adult asthma in the Agricultural Lung Health Study. *Eur Respir J* **56** (2020). |
| NLGN1 | 3q26.31 | **Smoking** | Drgon, T. et al. Genome-wide association for nicotine dependence and smoking cessation success in NIH research volunteers. *Mol Med* **15**, 21–7 (2009). |
| NAALADL2 | 3q26.31 | *Suggestive association with severe asthma exacerbations* | Herrera-Luis E et al. Genome-wide association study reveals a novel locus for asthma with severe exacerbations in diverse populations. Pediatr Allergy Immunol. 2021;32(1):106–115. |
| SV2C | 5q13.3 | **BMI**, diastolic **blood pressure** | GeneCards |
| PDE10A | 6q27 | Birth weight, **asthma**, and **BMI** | Melen, E. et al. Analyses of shared genetic factors between asthma and obesity in children. J Allergy Clin Immunol **126**, 631–7 e1-8 (2010). |
| MAGI2 | 7q21.11 | **Allergic diseases and atopy** | Freidin, M.B. et al. [Genome-wide association study of allergic diseases in Russians of Western Siberia]. Mol Biol (Mosk) **45**, 464–72 (2011). |
| MAGI2 | 7q21.11 | **Smoking** | Quach, B.C. et al. Expanding the genetic architecture of nicotine dependence and its shared genetics with multiple traits. *Nat Commun* **11**, 5562 (2020). |
| MAGI2 | 7q21.11 | **BMI** | GeneCards |
| MAGI2 | 7q21.11 | **Airway wall thickness** | GWAS Catalog |
| C9orf24 | 9p13.3 | **Airway repair** | Yoshisue, H. et al. Characterisation of ciliated bronchial epithelium 1, a ciliated cell-associated gene induced during mucociliary differentiation. *Am J Respir Cell Mol Biol* **31**, 491–500 (2004). |
| GUCY1A2 | 11q22.3 | Systolic/diastolic **blood pressure** | GeneCards |
| PRR20A/B/C/D/E | 13q21.1 | Systolic **blood pressure** | GeneCards |
| SEMA6D | 15q21.1 | **Smoking** | Minica, C.C. et al. Pathways to smoking behaviours: biological insights from the Tobacco and Genetics Consortium meta-analysis. Mol Psychiatry 22, 82–88 (2017). |

**Appendix 4—table 3.** References to previous relevant associated traits for each region/gene identified in early-onset mid-childhood remitting wheezing.

**Early-onset mid-childhood remitting wheezing**

| Gene(s) | Locus | Previous associated trait | Reference |
|---|---|---|---|
| CADM3, FCER1A, MPTX1, OR10J1 | 1q23.2 | **Neutrophil count**, CRP | Barreto, M. et al. Duffy phenotyping and FY*B-67T/C genotyping as screening test for benign constitutional neutropenia. *Hematol Transfus Cell Ther* (2020). |
| MRPL50P1 | 2p22.3 | **PM 2.5** exposure level and global **DNA methylation** level | Liu, J. et al. Genetic variants, PM2.5 exposure level and global DNA methylation level: A multi-center population-based study in Chinese. *Toxicol Lett* **269**, 77–82 (2017). |
| RAD18 | 3p25.3 | **Atopy/SPT** | Bouzigon, E. et al. Meta-analysis of 20 genome-wide linkage studies evidenced new regions linked to asthma and atopy. *Eur J Hum Genet* **18**, 700–6 (2010). |
| MRPL50P1 | 3q29 | **BMI** | Kettunen, J. et al. Multicenter dizygotic twin cohort study confirms two linkage susceptibility loci for body mass index at 3q29 and 7q36 and identifies three further potential novel loci. *Int J Obes (Lond)* **33**, 1235–42 (2009). |
| LSG1 | 3q29 | **BMI, eosinophil, and neutrophil count** | GeneCards |
| TMEM44-AS1, TMEM44, ATP13A3 | 3q29 | **Diastolic blood pressure** | GeneCards |
| SLC9B1 | 4q24 | **Eosinophil count** | Aschard, H. et al. Sex-specific effect of IL9 polymorphisms on lung function and polysensitization. *Genes Immun* **10**, 559–65 (2009). |
| SLC9B1 | 4q24 | **Allergic rhinitis** | Haagerup, A. et al. Allergic rhinitis--a total genome-scan for susceptibility genes suggests a locus on chromosome 4q24-q27. *Eur J Hum Genet* **9**, 945–52 (2001). |
| AP3B1 | 5q14.1 | **Vital capacity, BMI** | GeneCards, GWAS Catalog |
| RNU6-67P, SLITRK1, VENTXP2, UBE2D3P4, MTND4P1 | 13q31.1 | RNU6-67P/ rs976078: **food allergy** | Liu, X. et al. Genome-wide association study of maternal genetic effects and parent-of-origin effects on food allergy. *Medicine (Baltimore)* **97**, e0043 (2018). |
| ZFYVE1 | 14q24.2 | LDL **cholesterol** and systolic blood pressure | GWAS Catalog |
| PLCB4 | 20p12.3-p12.2 | **Neutrophil count** | Okada, Y. et al. Common variations in PSMD3-CSF3 and PLCB4 are associated with neutrophil count. *Hum Mol Genet* **19**, 2079–85 (2010). |

**Appendix 4—table 4.** References to previous relevant associated traits for each region/gene identified in late-onset wheezing.

**Late-onset wheezing**

| Gene(s) | Locus | Previous associated trait | Reference |
|---|---|---|---|
| KLHDC7A | 1p36.13 | 1p36.13: **metabolic syndrome** | Hoffmann, K. et al. A German genome-wide linkage scan for type 2 diabetes supports the existence of a metabolic syndrome locus on chromosome 1p36.13 and a type 2 diabetes locus on chromosome 16p12.2. *Diabetologia* **50**, 1418–22 (2007). |
| PPAP2B, PRKAA2 | 1p32.2 | PRKAA2: lymphocyte count and **asthma** susceptibility | Cusanovich, D.A. et al. The combination of a genome-wide association study of lymphocyte count and analysis of gene expression data reveals novel asthma candidate genes. *Hum Mol Genet* **21**, 2111–23 (2012). |
| HMGB1P18 | 1p31.1 | **Smoking, BMI** | GeneCards |
| ACOXL, BUB1 | 2q13 | ACOXL: **later onset asthma and obesity** | Zhu, Z. et al. Shared genetic and experimental links between obesity-related traits and asthma subtypes in UK Biobank. *J Allergy Clin Immunol* **145**, 537–549 (2020). |
| AMMECR1L | 2q14.3 | **Body height**, blood protein, growth, bone, and heart alterations | Moyses-Oliveira, M. et al. Inactivation of AMMECR1 is associated with growth, bone, and heart alterations. *Hum Mutat* **39**, 281–291 (2018). |
| RARB | 3p24.2 | **FEV$_1$/FVC**, adult lung function | Collins, S.A. et al. HHIP, HDAC4, NCR3 and RARB polymorphisms affect fetal, childhood and adult lung function. *Eur Respir J* **41**, 756–7 (2013). |
| KIAA2018, NAA50, SIDT1, CD200 | 3q13.2 | SIDT1: **FEV$_1$/FVC**, CD200: **adult-onset non-allergic asthma** | Siroux, V. et al. Genetic heterogeneity of asthma phenotypes identified by a clustering approach. *Eur Respir J* **43**, 439–52 (2014). |

*Appendix 4—table 4 Continued on next page*

*Appendix 4—table 4 Continued*

**Late-onset wheezing**

| Gene(s) | Locus | Previous associated trait | Reference |
|---|---|---|---|
| *TFDP2, XRN1* | 3q23 | XRN1: **eosinophil count**, 3q23: **allergic disease and atopic sensitisation** | Freidin, M.B. et al. [Genome-wide association study of allergic diseases in Russians of Western Siberia]. *Mol Biol (Mosk)* **45**, 464–72 (2011). |
| *SLAIN2, SLC10A4, FRYL* | 4p11 | FRYL: body height, age at menopause | GeneCards |
| *KRBA1, ZNF467* | 7q36.1 | Systolic blood pressure | GWAS Catalog |
| *GIMAP family, AOC1, LOC105375566* | 7q36.1 | AOC1: CV disease, **smoking**, GIMAP family: autoimmune diabetes, **asthma and allergy** | Heinonen, M.T. et al. GIMAP GTPase family genes: potential modifiers in autoimmune diabetes, asthma, and allergy. *J Immunol* **194**, 5885–94 (2015). |
| *PTPRN2* | 7q36.3 | **Eczema** | Bogari, N.M. et al. Whole exome sequencing detects novel variants in Saudi children diagnosed with eczema. *J Infect Public Health* **13**, 27–33 (2020). |
| *LOC105375631* | 8q21.3 | 8q21.3: **type 1 diabetes** | Mukhopadhyay, N., Noble, J.A., Govil, M., Marazita, M.L. & Greenberg, D.A. Identifying genetic risk loci for diabetic complications and showing evidence for heterogeneity of type 1 diabetes based on complications risk. *PLoS One* **13**, e0192696 (2018). |
| *NFIB, ZDHHC21* | 9p22.3 | 9p22.3: **asthma** (mean age <16 years) | Denham, S. et al. Meta-analysis of genome-wide linkage studies of asthma and related traits. *Respir Res* **9**, 38 (2008). |
| *SLC16A12, IFIT family, PANK1* | 10q23.31 | SLC16A12: body height, PANK1: **insulin resistance** | Yang, L. et al. P53/PANK1/miR-107 signalling pathway spans the gap between metabolic reprogramming and insulin resistance induced by high-fat diet. *J Cell Mol Med* **24**, 3611–3624 (2020). |
| *CBL, CCDC84, MCAM* | 11q23.3 | CBL: **sex hormone-binding globulin levels**; MCAM: blood protein levels | GWAS Catalog |
| *SPG11, CTDSPL2* | 15q15.3-q21.1 | CTDSPL2: **alcohol drinking** | GWAS Catalog |
| *ADAMTSL3, GOLGA6L4, UBE2Q2P8* | 15q25.2 | ADAMTSL3: **FEV$_1$/FVC**, lean mass | Karasik, D. et al. Disentangling the genetics of lean mass. *Am J Clin Nutr* **109**, 276–287 (2019). |
| *SSTR5-AS1, CACNA1H* | 16p13.3 | CACNA1H: **eosinophil count** | GWAS Catalog |
| *GSG1L* | 16p12.1 | 16p12.1: current **asthma** and rhino-conjunctivitis at 10–15 years | Sottile, G. et al. An association analysis to identify genetic variants linked to asthma and rhino-conjunctivitis in a cohort of Sicilian children. *Ital J Pediatr* **45**, 16 (2019). |
| *FAM19A5 or TAFA5* | 22q13.32 | **Obesity and metabolic dysfunction** | Recinella L. et al. Adipokines: New Potential Therapeutic Target for Obesity and Metabolic, Rheumatic, and Cardiovascular Diseases. Front Physiol. 2020 Oct 30;11:578966 |

## Appendix 5

### Replication of *ANXA1* top hits in PIAMA cohort

#### PIAMA cohort description

PIAMA (Prevention and Incidence of Asthma and Mite Allergy) is an ongoing birth cohort study. Details of the study design have been published previously (*Brunekreef et al., 2002*; *Wijga et al., 2014*). In brief, pregnant women were recruited from the general population through antenatal clinics in the north, west, and centre of the Netherlands in 1996–1997. The baseline study population consisted of 3963 newborns. Questionnaires were completed by the parents during pregnancy when the child was 3 months old, and then annually from 1 up to 8 years; at ages 11, 14, and 17 years, questionnaires were completed by the parents as well as the participants themselves.

#### LCA wheezing phenotypes

A six-class LCA model was identified including 3832 individuals with at least two observations of wheeze between 1 and 11–12 years of age. The identified classes were labelled: never/infrequent (2909, 75.91%), pre-school onset remitting (571, 14.90%), mid-childhood school remitting (108, 2.82%), intermediate onset remitting (106, 2.77%), school-age onset persisting (74, 1.93%), and continuous wheeze (64, 1.67%).

#### Replication analyses

We analysed associations between SNPs downstream of *ANXA1* (*Appendix 5—table 1*, *Appendix 5—figure 1*) and continuous wheezing in PIAMA, using the never/infrequent wheezing as the baseline category. Analyses were carried out in SPSS using a logistic regression model.

**Appendix 5—table 1.** Single nucleotide polymorphisms (SNPs) near *ANXA1* associated with persistent wheeze.

| Chr | Rsid | Position | A1 | A2 | freqA2 | Beta | SE | p-Value | Direction (3 GWAS) |
|-----|------|----------|-----|-----|--------|------|-----|---------|--------------------|
| 9 | rs75260654 | 75788108 | t | c | 0.02 | 0.90 | 0.16 | 6.66e-09 | --- |
| 9 | rs116849664 | 75820902 | t | c | 0.02 | 0.89 | 0.16 | 1.99e-08 | --- |
| 9 | rs78320984 | 75844302 | t | g | 0.02 | 0.81 | 0.15 | 6.41e-08 | --- |

A1 is the effect allele, A2 is the reference allele.

**Appendix 5—table 2.** Allele frequencies of *rs75260654* across different wheeze phenotypes.

| Phenotype | CC | CT | TT |
|-----------|-----|-----|-----|
| Never/infrequent | 5641 (97.2) | 161 (2.8) | 0 (0) |
| Early-onset pre-school remitting | 1409 (97.1) | 42 (2.9) | 0 (0) |
| Early-onset mid-childhood remitting | 572 (96.1) | 23 (3.9) | 0 (0) |
| Late-onset | 613 (95.2) | 31 (4.8) | 0 (0) |
| Early-onset persistent | 867 (94.2) | 51 (5.5) | 2 (0.2) |

**Appendix 5—table 3.** Selected immune eQTLs of rs75260654.

| Rsid | p-Value | Beta | SE | an | Symbol | Study |
|------|---------|------|-----|-----|--------|-------|
| rs75260654 | 0.014 | –0.65 | 0.26 | 382 | *ANXA1* | Quach_2016_monocyte_R848 |
| rs75260654 | 0.015 | –1.02 | 0.41 | 396 | *ANXA1* | Quach_2016_monocyte_IAV |

**Appendix 5—table 4.** Lung eQTLs of rs75260654.

| Rsid | p-Value | Beta | SE | an | Symbol | Study |
|------|---------|------|-----|-----|--------|-------|
| rs116849664 | 0.0489 | 0.22 | 0.11 | 620 | *ANXA1* | GTEx_exon_lung |
| rs78320984 | 0.0489 | 0.22 | 0.11 | 620 | *ANXA1* | GTEx_exon_lung |

**Appendix 5—table 5.** Functional enrichment for *ANXA1*: top 10 GO terms.

| Term name | Description | FDR value |
|---|---|---|
| GO.0007186 | G protein-coupled receptor signalling pathway | $3.57 \times 10^{-18}$ |
| GO.0006954 | Inflammatory response | $1.13 \times 10^{-16}$ |
| GO.0006874 | Cellular calcium ion homeostasis | $5.55 \times 10^{-15}$ |
| GO.0007204 | Positive regulation of cytosolic calcium ion concentration | $1.65 \times 10^{-14}$ |
| GO.0060326 | Cell chemotaxis | $1.95 \times 10^{-14}$ |
| GO.0006955 | Immune response | $4.23 \times 10^{-14}$ |
| GO.0006935 | Chemotaxis | $4.93 \times 10^{-14}$ |
| GO.0006952 | Defense response | $1.68 \times 10^{-13}$ |
| GO.0050801 | Ion homeostasis | $2.23 \times 10^{-13}$ |
| GO.0002376 | Immune system process | $2.87 \times 10^{-13}$ |

**Appendix 5—table 6.** Replication of associations between single nucleotide polymorphisms (SNPs) downstream of *ANXA1* and early-onset persistent wheezing in Prevention and Incidence of Asthma and Mite Allergy (PIAMA).

| | | | | CW (40) vs NI (1557) | | |
|---|---|---|---|---|---|---|
| Rsid | chr:position | R2 | A2/freqA2 | Beta | SE | p-Value |
| rs75260654 | 9:75788108 | 0.60 | c/0.02 | −0.287 | 0.91 | 0.75 |
| rs116849664 | 9:75820902 | 0.61 | c/0.02 | 0.119 | 1.08 | 0.91 |
| rs78320984 | 9:75844302 | 0.59 | g/0.02 | 0.125 | 1.04 | 0.90 |

A2 is the reference allele. CW = continuous wheezing, IR = intermediate wheezing derived from LCA 1–12 years in PIAMA.

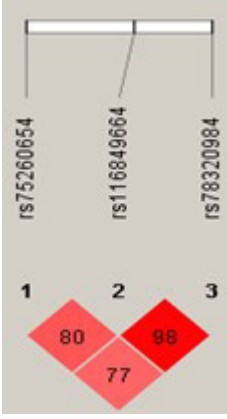

**Appendix 5—figure 1.** Linkage disequilibrium between single nucleotide polymorphisms (SNPs) downstream of *ANXA1* that were associated with persistent wheeze.

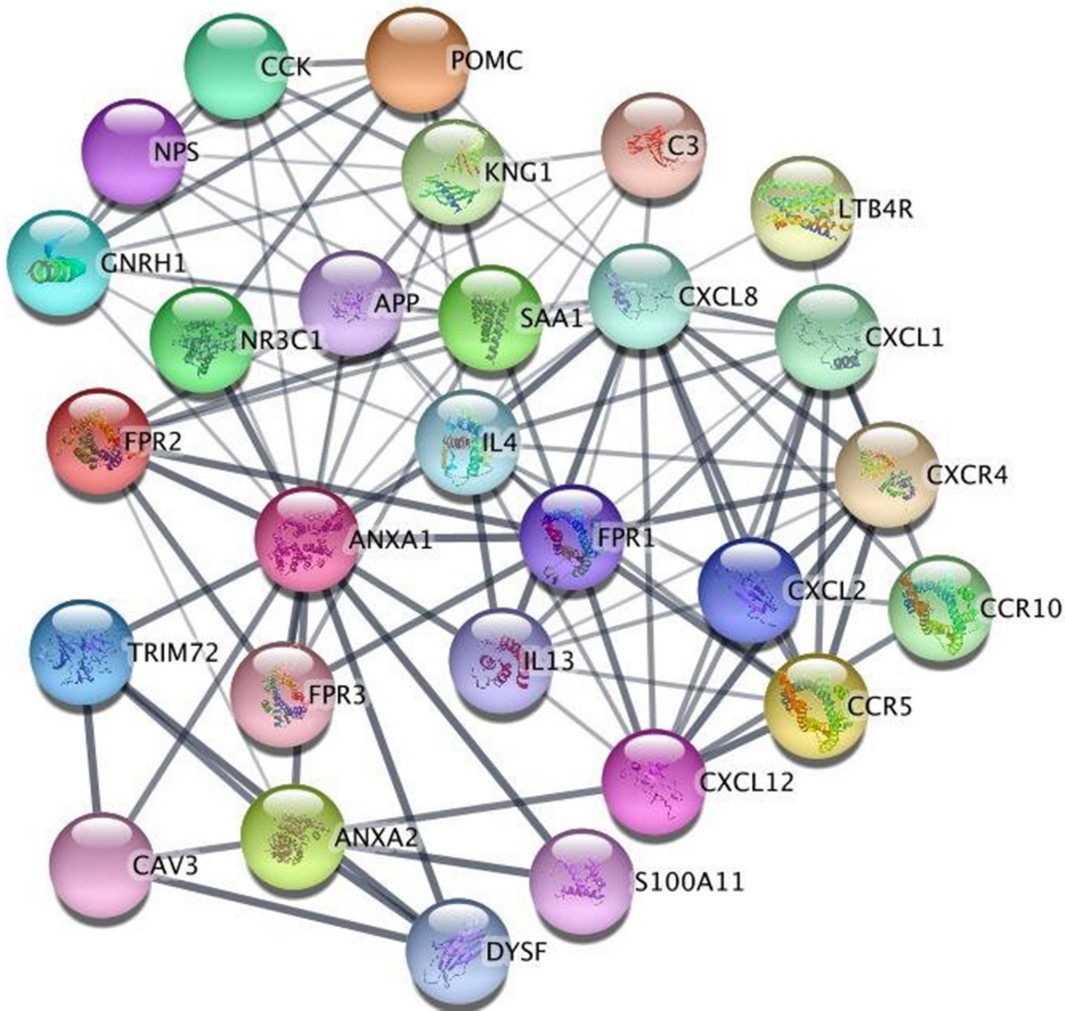

**Appendix 5—figure 2.** ANXA1 interactors. Protein-protein interaction of *ANXA1* including *IL-4, IL-13,* and *NR3C1.*

## Appendix 6

### Functional mouse experiments

#### Mice
In accordance with the Animals (scientific procedures) act 1986, all animal experiments were conducted under the approved UK Home Office Project License No: PPL 70/7643, reviewed by Imperial College's Animal Welfare and Ethical Review body. Female WT BALB/c and annexin A1 knock-out mice were purchased from Charles River (Bicester, UK). Animals aged 6–8 weeks of age received 25 µg intranasal instillation of either HDM (Greer Laboratories, Lenoir, NC, USA; Cat: XPB70D3A25) or PBS 3× a week for 3 weeks. Mice were sacrificed 4 hr post-final HDM challenge. Mice were housed under specific pathogen-free conditions and a 12:12 light:dark cycle. Food and water were supplied ad libitum. All animal experiments were completed twice, with N=4–6 per group.

#### Airway hyperresponsiveness
Airway hyperreactivity was measured using Flexivent. Lung resistance was measured in response to increasing doses of methacholine (3–100 mg/ml, Sigma, Poole, UK, Cat: A2251) in tracheotomised anaesthetised mice using an EMMS system (Electro-Medical Measurement Systems, UK).

#### Flow cytometry analysis
Bronchoalveolar lavage (BAL) was collected. BAL cells were restimulated with ionomycin and phorbol 12-myristate 13-acetate in the presence of brefeldin (Sigma), as previously described (*Branchett et al., 2021*). Specific antibodies for T1/ST2 staining were purchased from Morwell Diagnostics (Zurich, Switzerland). Cells were also stained for lineage negative cocktail, Ly6G, CD45, CD11b, CD11c, SiglecF. Labelled cells were acquired on a BD Fortessa (BD Biosciences, Oxford, UK) and analysed using FlowJo software (Treestar, Ashland, OR, USA). Details of antibodies used can be found in *Appendix 6—table 1*.

**Appendix 6—table 1.** Antibodies used in the flow cytometry analysis.

| Molecule | Manufacturer | Isotype | Conjugated dye | Clone |
|---|---|---|---|---|
| T1/ST2 | Morwell Diagnostics GMBH, Switzerland | Rat IgG1 | FITC | *DJ8* |
| CD45 | e-Bioscience Ltd, Hatfield, UK | Rat IgG2b | PerCP-CY5.5 | 30-F11 |
| CD11b | BD Biosciences, Oxford, UK | Rat IgG2b | e450 | M1/70 |
| CD11c | e-Bioscience Ltd, Hatfield, UK | Hamster IgG1 | PerCP-CY5.5 | N418 |
| Siglec F | BD Biosciences, Oxford, UK | Rat IgG2a | PE | E50-2440 |

#### Analysis of cytokines and chemokines
Murine lung tissue homogenate supernatants were processed as previously described (*Branchett et al., 2021*). Cytokine levels were analysed by ELISA: IL-4, IL-5 (PharMingen, Oxford, UK), IL-13 Ready-Set-Go kits (eBioscience).

#### Real-time PCR
Total RNA was extracted from murine lung tissue using an RNeasy Mini Kit (QIAGEN). Total RNA (1 µg) was reverse transcribed into cDNA using a High-Capacity cDNA Reverse Transcription Kit (Life Technologies, UK). Real-time PCRs were performed using TaqMan Gene Expression Master Mix and TaqMan Gene Expression probes, annexin A1, and HPRT (Applied Biosystems). Values were normalised to HPRT and gene expression was analysed using the change-in-threshold $2-\Delta CT$ method.

#### Annexin A1 immunohistochemistry
Paraffin-embedded mouse lung sections were stained with annexin A1 (R&D Systems, MAB3770). Annexin A1 primary antibody was followed by a secondary detection antibody (donkey anti-goat 488, Thermo Fisher, A11055). Annexin A1+ cells were quantified by manual counting under microscope and numbers averaged over four fields, from five biological replicates per group.

## Statistical analysis

Data are expressed as median±IQR. Statistical differences between groups were calculated using Mann-Whitney U test, unless otherwise specified. p-Values are indicated in figures.

