## [Editor Report]

The study uses a novel meta-analysis approach coupled with endotype discovery in GWAS studies of childhood wheezing to identify ANXA1 as a susceptibility locus. Functional data strengthens the conclusions from a relatively small sample size by GWAS standards. This is representative of a way forward for efficient genetic discovery in deeply phenotyped complex diseases, where there is much unrecognised heterogeneity.

---

## [Decision Letter]

**Decision letter after peer review:**

Thank you for submitting your article "A meta-analysis of genome-wide association studies of childhood wheezing phenotypes identifies ANXA1 as a susceptibility locus for persistent wheezing" for consideration by *eLife*. Your article has been reviewed by 3 peer reviewers, including Anurag Agrawal as the Reviewing Editor and Reviewer #1, and the evaluation has been overseen by Molly Przeworski as the Senior Editor.

Essential revisions:

1) Please provide more details on the meta-analysis strategy and rationale

Consider the other comments for greater clarity of the work

*Reviewer #1 (Recommendations for the authors):*

The work is interesting, novel, and technically rigorous. Some suggestions follow regarding the experimental mouse model part of the manuscript, which needs further editing for clarity and standardisation of terminology.

1. Figure 4A, font size needs to be increased.

2. Line 289-291 needs correction. 4E shows methacholine-induced bronchoconstriction, not allergen-induced reduced lung function.

3. Was baseline resistance (pre-methacholine) different? Any differences in the slope of the curve?

4. Airway remodelling changes should be described if possible.

*Reviewer #2 (Recommendations for the authors):*

Even though the study described by Granell et al. is an impressive piece of work with robust sets of evidence that support the derived conclusions, this reviewer has some suggestions the authors might want to consider:

1. A more detailed description of the association models applied to test the association with wheezing phenotypes, including the covariates used, is encouraged.

2. Which approach was used for the meta-analysis? Did you take into account the heterogeneity among studies to choose fixed or random-effect models to be applied?

3. Could you please provide an explanation of the genomic inflation found for the persistent wheezing phenotype (Figure E1)?

4. Table E4: Could you please explain the reason for inclusion/exclusion?

*Reviewer #3 (Recommendations for the authors):*

1. Regarding the definition of wheeze phenotypes from infancy to adolescence, were transitions across different phenotypes over time possible? Was time-varying phenotype of concern? Did the authors evaluate the accuracy of phenotyping using metrics such as sensitivity, specificity, and positive and negative predictive values? And how much of the variation in each wheeze phenotype can be explained by genetic profile?

2. In order to identify primary and secondary independent loci, can the authors perform the conditional analysis? Further, considering the LD structure, credible set analysis, and colocalization analysis can help identify potential causal variants.

3. Can the authors provide further explanations on the biology using pQTL and TWAS?

4. Can the authors use DEPICT or other tools to identify relevant tissues/cell types that are highly expressed?

5. The heterogeneity in the genetic profile for the wheeze phenotypes can be better illustrated, such as by using a scatter plot to show the differences in genetic effect sizes. A heterogeneity p-value can be computed. In addition, what was the genetic correlation between the wheeze phenotypes?

---

## [Author Response]

Essential revisions:1) Please provide more details on the meta-analysis strategy and rationale

Please see Reviewer 2 Q2.

Consider the other comments for greater clarity of the workReviewer #1 (Recommendations for the authors):The work is interesting, novel, and technically rigorous. Some suggestions follow regarding the experimental mouse model part of the manuscript, which needs further editing for clarity and standardisation of terminology.1. Figure 4A, font size needs to be increased.

Figure 5A (4Ain previous version) has been updated.

2. Line 289-291 needs correction. 4E shows methacholine-induced bronchoconstriction, not allergen-induced reduced lung function.

We have corrected this in line 322 of the Revised Manuscript.

3. Was baseline resistance (pre-methacholine) different? Any differences in the slope of the curve?

No statistically significant difference was observed in resistance at baseline. However, a leftward shift in the curve was observed.

4. Airway remodelling changes should be described if possible.

We did not assess changes in remodelling in these experiments. Here, we utilised a 3-week HDM challenge model, whereas to fully characterise changes in airways remodelling, mice would need to be assessed later than 3 weeks post HDM.

Reviewer #2 (Recommendations for the authors):Even though the study described by Granell et al. is an impressive piece of work with robust sets of evidence that support the derived conclusions, this reviewer has some suggestions the authors might want to consider:1. A more detailed description of the association models applied to test the association with wheezing phenotypes, including the covariates used, is encouraged.

We agree with this reviewer’s comment, and we have made this clearer in the Methods section on lines 185-7 Revised Manuscript:

“We used SNPTEST v2.5.2(26) with a frequentist additive multinomial logistic regression model (-method newml), using the never/infrequent wheeze as the reference and without including any covariates.”

2. Which approach was used for the meta-analysis? Did you take into account the heterogeneity among studies to choose fixed or random-effect models to be applied?

These details were already available in the Supplement, and they can now be found in Appendix 2: GWAS Meta-analysis:

“We used the option SCHEME STDERR in METAL to implement an effect-size based method weighted by each study-specific standard error in a fixed-effects model.”

3. Could you please provide an explanation of the genomic inflation found for the persistent wheezing phenotype (Figure E1)?

We would interpret Figure 1—figure supplement 1 (previously Figure E1) as an association between SNPs and Persistent wheeze rather than inflation. For instance, we did not observe any deviation in the other phenotypes. We now report the phenotype specific genomic inflation factor for each wheezing phenotype in lines 222-5 Revised Manuscript:

“We observed slight deflation of the meta-analysis pvalues in our summary statistics. Genomic Inflation Factor (λ) for Early-onset Pre-school Remitting=0.96, Early-onset Mid-childhood Remitting = 0.94, Late-onset=0.96, and Early-onset Persistent wheezing=0.97.”

And have added this text in Appendix 2: Genetic Control:

“The genomic inflation factor (λ) was calculated using the scipy.stats.chi2 module in Python. The chi-squared test statistics from the meta-analysis p-values were first obtained. Then the observed median chi-squared statistic from the calculated chi-squared test statistics were calculated. Finally, the genomic inflation factor (λ) was derived by dividing the observed median chi-squared statistic by the expected median chi-squared statistic.”

4. Table E4: Could you please explain the reason for inclusion/exclusion?

Thank you, this was not properly explained in the main text. Individuals with missing genetic data, as well as related and non-European individuals were excluded. We have now made this clear in line 212-4 Revised Manuscript:

“Individuals with missing genetic data, as well as related and non-European individuals were excluded. Comparison of included vs. excluded individuals across cohorts (per cohort and time point) is shown in Appendix 1 and Appendix 1-Table 4.”

Reviewer #3 (Recommendations for the authors):1. Regarding the definition of wheeze phenotypes from infancy to adolescence, were transitions across different phenotypes over time possible? Was time-varying phenotype of concern? Did the authors evaluate the accuracy of phenotyping using metrics such as sensitivity, specificity, and positive and negative predictive values?

No, LCA does not allow transitions across different phenotypes over time.

Was time-varying phenotype of concern? Did the authors evaluate the accuracy of phenotyping using metrics such as sensitivity, specificity, and positive and negative predictive values?

We have explored time-varying phenotype in the context of LCA in our recent paper (Haider S et al. Modeling Wheezing Spells Identifies Phenotypes with Different Outcomes and Genetic Associates. Am J Respir Crit Care Med. 2022 Apr 15;205(8):883-893. doi: 10.1164/rccm.202108-1821OC. PMID: 35050846; PMCID: PMC9838626.) where we compared the robustness of LCA and PAM phenotypes. In this paper we show that latent class analysis- and Partition Around Medoids (PAM) clustering of spell-based phenotypes appeared similar, but within-phenotype individual trajectories and phenotype allocation differed substantially. The spell-based approach was much more robust in dealing with missing data, and the derived clusters were more stable and internally homogeneous than those derived by LCA. If anything, any instability of LCA phenotypes over time would reduce the power of the current study, yet we still identified genome-wide significant associations unique to specific wheeze clusters. None-the-less future studies should explore the utilisation of PAM derived phenotypes in genetic analysis of allergic disease.

And how much of the variation in each wheeze phenotype can be explained by genetic profile?

We have not explored this question in this paper, but we will do so in our follow-up paper.

2. In order to identify primary and secondary independent loci, can the authors perform the conditional analysis? Further, considering the LD structure, credible set analysis, and colocalization analysis can help identify potential causal variants.3. Can the authors provide further explanations on the biology using pQTL and TWAS?4. Can the authors use DEPICT or other tools to identify relevant tissues/cell types that are highly expressed?

The reviewer makes a number of helpful suggestions of additional post-GWAS analyses that could be undertaken on our results. Given the size of the manuscript and the extensive incorporation of mouse model data for the ANAX1 loci, we feel that adding further post GWAS analyses to the supplementary data is beyond the scope of this paper and we are currently working on a new follow-up paper that will focus on fine mapping of these loci.

5. The heterogeneity in the genetic profile for the wheeze phenotypes can be better illustrated, such as by using a scatter plot to show the differences in genetic effect sizes. A heterogeneity p-value can be computed.

We agree with the reviewer’s suggestion and have produced scatter plots to illustrate the heterogeneity in the genetic profile of the wheezing phenotypes. We have added these plots in Figure 2—figure supplement 1 and the following text has been added in lines 230-3 Revised Manuscript:

“Scatter plots in Figure 2—figure supplement 1 show the heterogeneity in the genetic profile of the wheeze phenotypes. The plots show that all signals were phenotype-specific at p<10-5 and only nominal associations were shared across wheezing phenotypes. More details on how these plots were derived can be found in Appendix 2: Heterogeneity scatter plots.”

In addition, what was the genetic correlation between the wheeze phenotypes?

We identified exclusive subgroups of SNPs associated with each wheezing phenotype, and only two locus were shared with Persistent and Early-onset Mid-childhood Remitting Wheezing (chr17q12 and chr17q21) therefore the genetic correlation between the wheeze phenotypes is likely to be small.